# Polarized NHE1 and SWELL1 regulate migration direction, efficiency and metastasis

Yuqi Zhang[1,2], Yizeng Li[3], Keyata N. Thompson[4], Konstantin Stoletov[5], Qinling Yuan[1,2], Kaustav Bera [1,2], Se Jong Lee[1,2], Runchen Zhao[1,2], Alexander Kiepas [1,2], Yao Wang [1,2], Panagiotis Mistriotis [1,2,6], Selma A. Serra[7], John D. Lewis [5], Miguel A. Valverde [7], Stuart S. Martin [4,8], Sean X. Sun [1,2,9,10] ✉ & Konstantinos Konstantopoulos [1,2,10,11] ✉

Cell migration regulates diverse (patho)physiological processes, including cancer metastasis. According to the Osmotic Engine Model, polarization of NHE1 at the leading edge of confined cells facilitates water uptake, cell protrusion and motility. The physiological relevance of the Osmotic Engine Model and the identity of molecules mediating cell rear shrinkage remain elusive. Here, we demonstrate that NHE1 and SWELL1 preferentially polarize at the cell leading and trailing edges, respectively, mediate cell volume regulation, cell dissemination from spheroids and confined migration. SWELL1 polarization confers migration direction and efficiency, as predicted mathematically and determined experimentally via optogenetic spatiotemporal regulation. Optogenetic RhoA activation at the cell front triggers SWELL1 re-distribution and migration direction reversal in SWELL1-expressing, but not SWELL1-knockdown, cells. Efficient cell reversal also requires Cdc42, which controls NHE1 repolarization. Dual NHE1/SWELL1 knockdown inhibits breast cancer cell extravasation and metastasis in vivo, thereby illustrating the physiological significance of the Osmotic Engine Model.

Cell migration is a pivotal step during the process of cancer metastasis, as it enables cancerous cells disseminating out of a primary tumor to move through tissues and ultimately develop metastatic colonies in distant organs. Metastasizing cells migrate either by remodeling their surrounding three-dimensional (3D) extracellular matrix (ECM) to open up migratory paths, by following leader cells such as tumor-associated stromal cells that generate such paths, or by migrating through pre-existing, 3D longitudinal channel-like tracks created by various anatomical structures[1-3].

It is well established that cell motility is governed by cell-matrix interactions and the actomyosin cytoskeleton. Ion channels and ion transporters have also been recognized as important constituents of the cell migration machinery[4,5]. Yet, our understanding of how and which of them regulate cell locomotion is, at best incomplete. As

[1]Department of Chemical and Biomolecular Engineering, The Johns Hopkins University, Baltimore, MD 21218, USA. [2]Johns Hopkins Institute for NanoBio-Technology, The Johns Hopkins University, Baltimore, MD 21218, USA. [3]Department of Biomedical Engineering, Binghamton University, SUNY, Binghamton, NY 13902, USA. [4]Marlene and Stewart Greenebaum National Cancer Institute Comprehensive Cancer Center, University of Maryland School of Medicine, Baltimore, MD 21201, USA. [5]Department of Oncology, University of Alberta, Edmonton, AB T6G 2E1, Canada. [6]Department of Chemical Engineering, Auburn University, Auburn, AL 36849, USA. [7]Laboratory of Molecular Physiology, Department of Experimental and Health Sciences, Universitat Pompeu Fabra, 08003 Barcelona, Spain. [8]Department of Physiology, University of Maryland School of Medicine, Baltimore, MD 21201, USA. [9]Department of Mechanical Engineering, The Johns Hopkins University, Baltimore, MD 21218, USA. [10]Department of Biomedical Engineering, The Johns Hopkins University, Baltimore, MD 21218, USA. [11]Department of Oncology, The Johns Hopkins University, Baltimore, MD 21205, USA. ✉e-mail: ssun@jhu.edu; konstant@jhu.edu

proposed by the Osmotic Engine Model (OEM)[6,7], cell locomotion in confined spaces is mediated by highly coordinated cycles of local isosmotic swelling at the cell leading edge and shrinkage at the trailing edge, driven by the polarization of select ion transporters, ion channels and aquaporins (AQPs). We previously reported that the Na$^+$/H$^+$ exchanger 1 (NHE1) together with AQP5 polarize at the leading edge of migrating cells in confined microenvironments[7]. NHE1 supports confined migration even after cell treatment with a high dose of latrunculin A (LatA), which completely disrupts F-actin[7]. However, the role of NHE1 in promoting isosmotic swelling has yet to be established. Furthermore, it is currently unknown which ion channel(s) and AQP(s) preferentially localize at the cell rear, and mediate isosmotic shrinkage of the cell trailing edge. Also, does the spatial polarization of these molecules confer migration directionality? Do they act in concert with NHE1 and AQP5 to mediate efficient migration in vitro? Lastly, do they affect breast cancer cell metastasis in vivo? To address these fundamental and translational questions, we combined microfluidics with live-cell imaging, novel optogenetic tools, mathematical modeling, and in vivo mouse and ex vivo chick embryo models.

Here, we show that SWELL1 (LRRC8A)[8,9] and AQP4 preferentially localize at the trailing edge of confined breast cancer cells and that SWELL1 mediates isosmotic shrinkage consistent with its role in regulating local volume decrease. By developing optogenetic tools to control the spatiotemporal pattern of SWELL1, we demonstrate that its polarization at the cell rear confers migration direction. We also developed a multi-phase, steady-state cell migration mathematical model to predict the relation between SWELL1 expression and cell migration. Furthermore, we delineate the effects of individual and dual knockdown of NHE1 and SWELL1 on cell dissemination from 3D spheroids in vitro as well as on breast cancer growth and metastasis using an orthotopic mouse model and an ex vivo chick embryo model.

## Results

### NHE1 and SWELL1 preferentially polarize at the cell leading and trailing edges, respectively, and mediate cell volume regulation and efficient confined migration

According to OEM[6,7], cells migrating inside confining channels display a spatial gradient of distinct ion transporters and AQPs in the cell membrane so that local swelling at the leading edge and shrinkage at the trailing edge, respectively, facilitate net cell locomotion. In line with previous findings in various tumor cell types[7], NHE1 (Fig. 1a, b and Supplementary Fig. 1a, b) is polarized at the cell leading edge of MDA-MB-231 breast cancer cells migrating inside polydimethylsiloxane (PDMS)-based confining channels of prescribed dimensions (Width = 3 μm; Height = 10 μm; Length = 200 μm) coated with collagen type I. Consistent with its role in cell protrusion[7], NHE1 mediates isosmotic cell swelling in confinement, as evidenced by the reduced volume of NHE1-silenced relative to scramble control (SC) MDA-MB-231 cells (Fig. 1c, d) measured from confocal 3D image reconstructions of Lifeact-GFP-labeled cells (Supplementary Fig. 1c)[10]. This finding was further validated by measuring the cell longitudinal area (Supplementary Fig. 1d), which serves as a proxy of cell volume since MDA-MB-231 cells contact all four channel walls inside a narrow channel[7,11] using different short hairpin (sh)RNA sequences (Supplementary Fig. 1e, f). In line with prior work[7], NHE1 knockdown suppresses confined migration (Fig. 1e and Supplementary Fig. 1g).

Because cell migration involves a cycle of isosmotic regulatory volume increase (RVI) at the front and regulatory volume decrease (RVD) at the rear[5,7], and confined cells present preferential localization of the RVI-mediating NHE1 at the leading edge, we next examined whether the RVD-mediating SWELL1 chloride channel and select AQPs to localize at the trailing edge. Indeed, live-cell imaging using ectopically expressed SWELL1-GFP (Fig. 1a, b and Supplementary Fig. 2a; Supplementary Movie 1) and AQP4-mCherry (Supplementary Fig. 2a, b; Supplementary Movie 2) reveals that they are polarized and

colocalized at the trailing edge of MDA-MB-231 cells migrating in confinement. Immunofluorescence assays using an anti-SWELL1 monoclonal antibody also confirmed the preferential enrichment of endogenous SWELL1 at the cell rear (Supplementary Fig. 2c, d). Whole-cell patch-clamp experiments reveal that SC MDA-MB-231 cells exhibit SWELL1-mediated chloride currents, $I_{Cl,vol}$, after exposure to hypotonicity, as SWELL1 silencing (Fig. 1c) nearly abolishes these currents (Supplementary Fig. 2e). A similar reduction in hypotonicity-induced chloride currents is observed using the selective SWELL1 inhibitor, DCPIB (37.5 μM), in SC cells (Supplementary Fig. 2e). DCPIB also exerts a modest inhibitory effect on $I_{Cl,vol}$ currents in SWELL1-knockdown (KD) cells (Supplementary Fig. 2e), suggesting the presence of a very small residual amount of SWELL1 in these cells. Importantly, SWELL1 mediates isosmotic cell shrinkage, as evidenced by testing SWELL1-KD cells generated with different shRNA sequences (Fig. 1c and Supplementary Fig. 1e), which exhibit increased volume (Fig. 1d) and larger longitudinal area (Supplementary Figs. 1d and 2f) but reduced motility (Fig. 1e and Supplementary Fig. 2g) relative to SC cells in confinement. In view of the colocalization of SWELL1 and AQP4 at the cell rear and because AQP4-KD (Supplementary Fig. 2h) compared to SC cells also exhibit increased longitudinal area (Supplementary Fig. 2i), we postulate that SWELL1 works in concert with AQP4 to mediate shrinkage of the cell rear. Along these lines, AQP4 silencing suppresses confined migration (Supplementary Fig. 2j). Dual knockdown of NHE1 and SWELL1 does not alter cell volume (Fig. 1d) or longitudinal area (Supplementary Fig. 1d) in confinement, which is in accord with their individual counteracting effects on cell volume regulation.

In line with the role of NHE1 and SWELL1 in isosmotic swelling and shrinkage, respectively, their individual knockdown impaired MDA-MB-231 cell entry and migration into confining channels (Fig. 1e, f). Importantly, dual silencing of NHE1 and SWELL1 results in a cooperative and pronounced inhibition of cell entry and confined migration (Fig. 1e, f). Of note, NHE1 or/and SWELL1 do not affect the proliferation rate of MDA-MB-231 cells (Supplementary Fig. 2k). To provide further support for the critical involvement of NHE1 and SWELL1 in confined migration, we demonstrate that cell velocities in Na$^+$ free and Cl$^-$ low solutions are reduced relative to appropriate control media, and mirror those of NHE1- and SWELL1-KD cells, respectively (Supplementary Fig. 3a). To extend our findings beyond the MDA-MB-231 cell model, we demonstrate that NHE1 and SWELL1 preferentially polarize at the cell front and rear, respectively, of metastatic SUM159 breast cancer cells, migrating in confinement[12] and mediate isosmotic swelling and shrinkage (Supplementary Fig. 3b, c). Moreover, pharmacological inhibition of NHE1 by EIPA (40 μM)[5,7] and/or SWELL1 by DCPIB (40 μM)[8] markedly suppresses the migration of SUM159 cells and metastatic PTEN$^{-/-}$/KRAS(G12V) MCF-10A cells[12,13], which bear a double mutation that results in PTEN loss and overexpression of activated KRAS(G12V) (Supplementary Fig. 3d,e). Cells adjust their volume by transporting primarily Na$^+$, Cl$^-$, K$^+$ via plasma membrane channels and transporters[4]. Although they possess several Na$^+$, Cl$^-$, and K$^+$ transporters, such as the Na$^+$/ K$^+$/2Cl$^-$ (NKCC) co-transporters, we have excluded the potential involvement of NKCC, as its pharmacological inhibition fails to alter the migration of scramble control or dual NHE1- and SWELL1-KD cells (Supplementary Fig. 3f).

To extend the physiological relevance of our results, we examined the functional roles of NHE1 and/or SWELL1 in cell dissemination from 3D breast cancer spheroids embedded in 3D collagen gels (Fig. 1g; Supplementary Movie 3) or on 2D collagen I-coated surfaces (Supplementary Movie 4). In concert with the findings inside narrow channels, dual depletion of NHE1 and SWELL1 was markedly more efficient than individual knockdowns in delaying MDA-MB-231 cell dissemination from spheroids and their subsequent migration inside 3D collagen gels (Fig. 1h–k). As another measure of local cell invasiveness in 3D collagen gels, we quantified the area and circularity of spheroids after having been embedded in 3D collagen gels for 12 h.

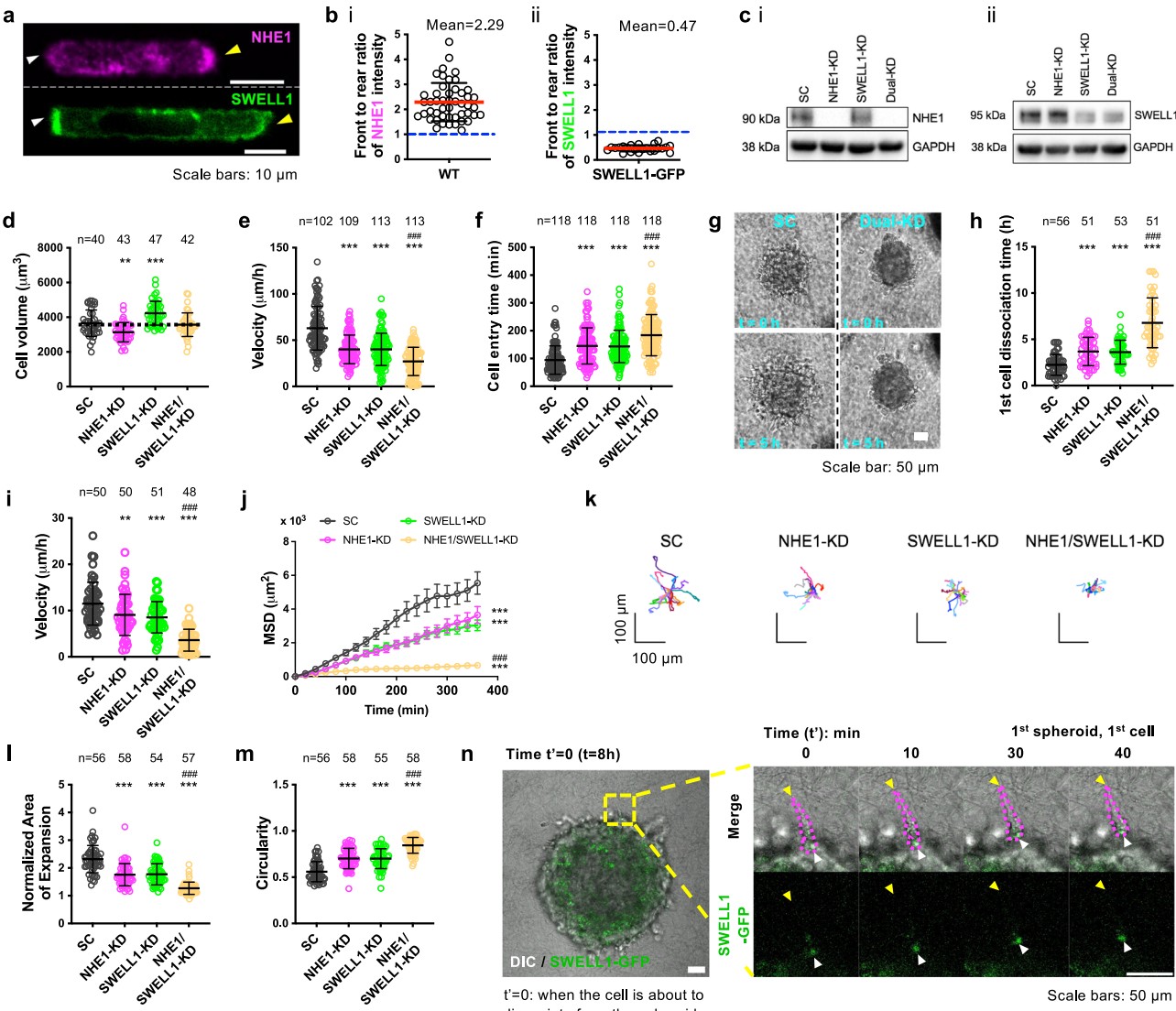

**Fig. 1 | Distinct spatial localization patterns of NHE1 and SWELL1, and their roles in cell volume regulation and confined migration. a**, Top*:* Image of a cell stained for NHE1 showing preferential localization at the leading edge (yellow arrowhead). Bottom*:* Image of another cell showing intense localization of SWELL1-GFP at the cell rear (white arrowhead). **b** Front to rear ratio of (i) endogenous NHE1 (*n* = 48) or (ii) SWELL1-GFP intensity (*n* = 23) in confined cells. Data represent mean ± SD from four independent experiments. **c** Western blots of cells transduced with SC or shRNA sequences against NHE1 and/or SWELL1. GAPDH served as a loading control. Uncropped blots in Source Data. **d** Effects of NHE1 and/or SWELL1 knockdown on cell volume inside confining channels. Data represent mean ± SD for cells analyzed from three independent experiments. **e, f** Effects of NHE1 and/or SWELL1 knockdown on **e** migration velocity and **f** cell entry time in confining channels. Data represent mean ± SD for cells analyzed from 3 independent experiments. **g** Images showing dissemination of SC and dual NHE1- and SWELL1-KD cells from spheroids embedded in 3D collagen gels at t = 0 and 5 h.

**h–k** Effects of NHE1 and/or SWELL1 knockdown on **h** the time for the first cell to dissociate from spheroids, and the **i** migration velocity, **j** mean squared displacement, and **k** trajectories of disseminated cells in 3D. Data represent mean ± SD for cells analyzed from three independent experiments. **l, m** Effects of NHE1 and/or SWELL1 knockdown on **l** normalized area of expansion at t = 12 h relative to t = 0 and **m** circularity of the spheroids at t = 0 embedded in 3D collagen gels. Data represent mean ± SD for cells analyzed from three independent experiments. **n** Time-lapse montage of a SWELL1-GFP-tagged cell (outlined by dashed magenta lines) dissociating from a spheroid embedded in collagen. White arrowheads denote SWELL1 polarization at the cell rear. Yellow arrowheads indicate the cell leading edge. **\*\****p* < 0.01 and *\*\*\*p* < 0.001 relative to SC, *###p* < 0.001 relative to either of single KD cells. Tests performed: **d, e, h, i, m** one-way ANOVA followed by Tukey's post hoc test, **f, l** Kruskal–Wallis followed by Dunn's, or **j** two-way ANOVA followed by Tukey's. The number of cells analyzed is indicated in each panel. Cell model: MDA-MB-231.

Spheroids consisting of dually depleted cells displayed a smaller area of expansion and increased circularity (Fig. 1l, m), indicative of a less invasive phenotype[14]. The roles of NHE1 and SWELL1 in cell dissemination from 3D spheroids and area of spheroid expansion were also verified using SUM159 cells (Supplementary Fig. 4a). To extend the physiological relevance of our findings, we further demonstrate that SWELL1 is preferentially polarized at the cell trailing edge during the dissociation of SWELL1-GFP-tagged cells from spheroids embedded in a 3D collagen gel (Fig. 1n and Supplementary Fig. 4b). Collectively, our data support a model by which the repeated and coordinated cycle of

local isosmotic swelling at the leading edge and shrinkage at the trailing edge mediated by NHE1 and SWELL1, respectively, supports migration in confinement as well as cell dissemination from tumor spheroids.

## SWELL1 polarization controls cell migration direction and efficiency, as predicted mathematically and determined experimentally via optogenetic spatiotemporal regulation

We developed a multi-phase model[15,16] to understand actin-water-ion-coupled cell migration in confinement. The model accounts for F-actin,

G-actin, cytosol (essentially water), charged ions, and focal adhesions, which provide force to the cell through the actin network. Myosin contraction is not explicitly modeled, and pressure in the actin network is dominated by passive pressure due to actin swelling. The intracellular governing equations for actin and cytosol velocities as well as boundary fluxes satisfy mass and force balances[15]. The boundary condition of the actin-network phase is linked to the rate of actin polymerization and depolymerization[15], and the boundary fluxes of the cytosol are linked to the water flux across the cell membrane. The actin-network phase remains within the cell while the cytosol phase exchanges with the extracellular medium. Water influx and efflux are determined by the total chemical potential difference across the cell membrane[7], i.e., $J_{water} = -\alpha(\Delta p - \Delta\Pi)$, where $\Delta p$ and $\Delta\Pi$ are the hydrostatic and osmotic pressure differences across the cell membrane, respectively. The hydrostatic pressure is obtained from the cytosol pressure. The osmotic pressure is determined by the total concentration of all the ionic species under consideration. The model also takes into account the transport of key ionic species across the cell membrane, including $Na^+$, $H^+$, and $Cl^-$, and assumes electroneutrality at equilibrium.

In light of experimental data revealing SWELL1 polarization at the cell rear of migrating cells in confinement (Fig. 1a, b and Supplementary Figs. 2a, c, d), we aimed to understand how SWELL1 spatial localization impacts migration. By altering the permeability coefficients of $Cl^-$ at the cell front ($\alpha^f_{Cl,p}$) and rear ($\alpha^b_{Cl,p}$), which depend not only on the SWELL1 channel property but also on the density of these channels in the membrane, the mathematical model predicts that maximal migration velocity is achieved when SWELL1 is enriched at the cell rear,

consistent with its role in RVD (Fig. 2a, Supplementary Fig. 5a). Importantly, equal distribution of SWELL1 expression at the cell poles is sufficient to cease motility, whereas preferential polarization at the cell front results in the reversal of migration direction, as evidenced by the negative velocity values (Fig. 2a, Supplementary Fig. 5a).

To test the model prediction and directly establish the role of SWELL1 polarization pattern in the direction and efficiency of confined migration, we developed optogenetic tools to regulate its spatiotemporal localization on the cell membrane, using the cryptochrome 2 (Cry2)-CIBN light-gated dimerizer system[17,18]. This technology relies on the fusion of SWELL1 to Cry2-mCherry (OptoSWELL1) and its GFP-labeled dimerization partner CIBN engineered to bind to the plasma membrane via the CAAX anchor (CAAX-CIBN-GFP) in response to blue light (Fig. 2b). Before light stimulation, SWELL1 localizes primarily at the trailing edge of cells migrating inside confining channels (Fig. 2b–d, Supplementary Movie 5). Light stimulation at the cell leading edge gradually promotes local SWELL1 enrichment, which is accompanied by a reduction of SWELL1 intensity at the opposite pole (Fig. 2b–d, Supplementary Movie 5). During this process, cell migration velocity decreases as the front-to-rear ratio of SWELL1 expression progressively increases (Fig. 2a–d). When SWELL1 is equally distributed at the cell poles at $t = t_1$, cell motility halts (Fig. 2c, d). Further light stimulation ($t > t_1$) induces preferential SWELL1 enrichment at the cell front along with the concomitant reversal of migration direction, as evidenced by the negative velocity values (Fig. 2b–d, Supplementary Movie 5). The relative fold change of front to rear SWELL1 intensity ratio for each cell following optogenetic stimulation is shown in Supplementary Fig. 5b.

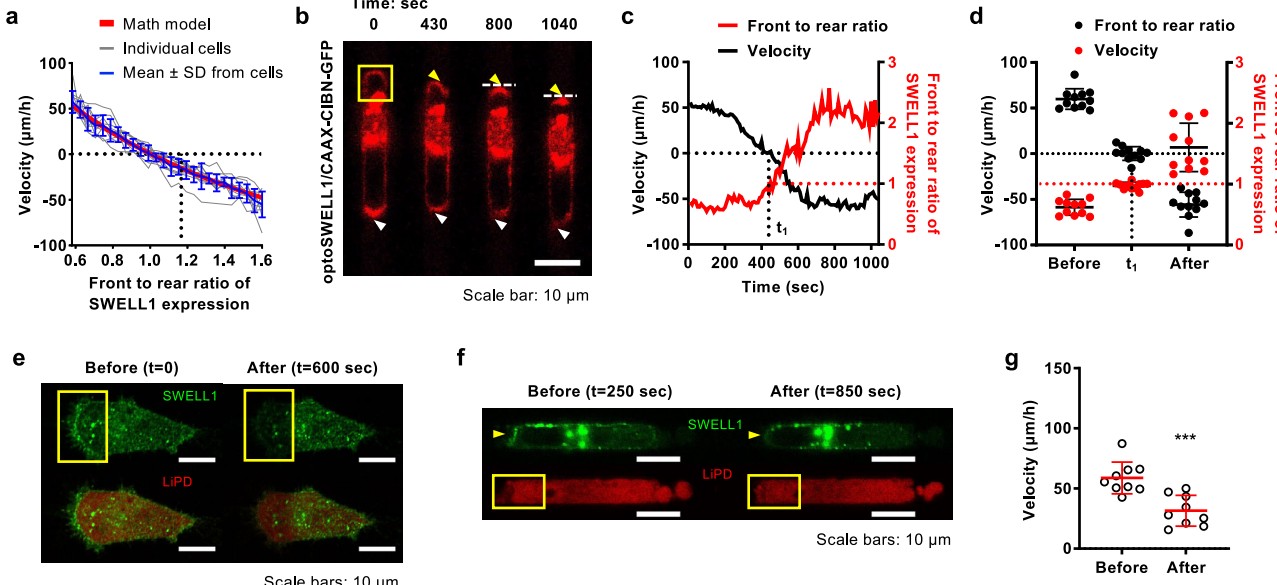

**Fig. 2 | SWELL1 polarization controls cell migration direction and efficiency. a** A multi-phase, steady-state mathematical model (shown in red), which accounts for actin-cytosol (water)-ions-coupled cell migration, was compared to experimental measurements from 11 individual cells subjected to optogenetic stimulation (shown in gray). Cells were obtained from three independent experiments. The model predicts that SWELL1 enrichment at the cell rear controls migration direction and velocity, which is in line with the mean ± SD of the experimental data (shown in blue). **b** Time-lapse montage of a representative MDA-MB-231 cell expressing OptoSWELL1 and CAAX-CIBN-GFP following light stimulation at the cell leading edge in a region enclosed by the yellow box. Gradual SWELL1 accumulation at the cell front (yellow arrowhead) is accompanied by reduction at the cell rear (white arrowhead). White dash lines denote the "new" trailing edge. **c** Instantaneous migration velocity (black line) and front to rear ratio of SWELL1 intensity (red line) of the cell shown in **b**. At t = 430 sec (t₁), SWELL1 is equally distributed at both cell

poles and its motility ceases. Further SWELL1 enrichment at the cell front results in reversal of migration direction as depicted by negative velocity values. **d** Migration velocity (black dots) and front to rear ratio of SWELL1 fluorescence intensity (red dots) of cells before optogenetic stimulation, at t = t₁ and t ≥ 1000 sec after stimulation. Data represent the mean ± SD for 11 cells from 3 independent experiments, also shown in **a**. **e**, **f** Montage of cells expressing SWELL1-GFP and LiPD system **e** on 2D and **f** inside confining channels. Merged images of SWELL1-GFP and LiPD system are shown in the lower panels (**e**). Light stimulation was applied in a region enclosed by yellow boxes just after **e** t = 0 or **f** t = 250 sec. Light-induced SWELL1 degradation is observed after 600 sec. **g** Migration velocity of cells expressing SWELL1-GFP and LiPD system before and after light stimulation. Data represent the mean ± SD for 9 cells from 4 independent experiments. ***p < 0.001 relative to before light stimulation. Significance was determined using two-tailed paired t-test.

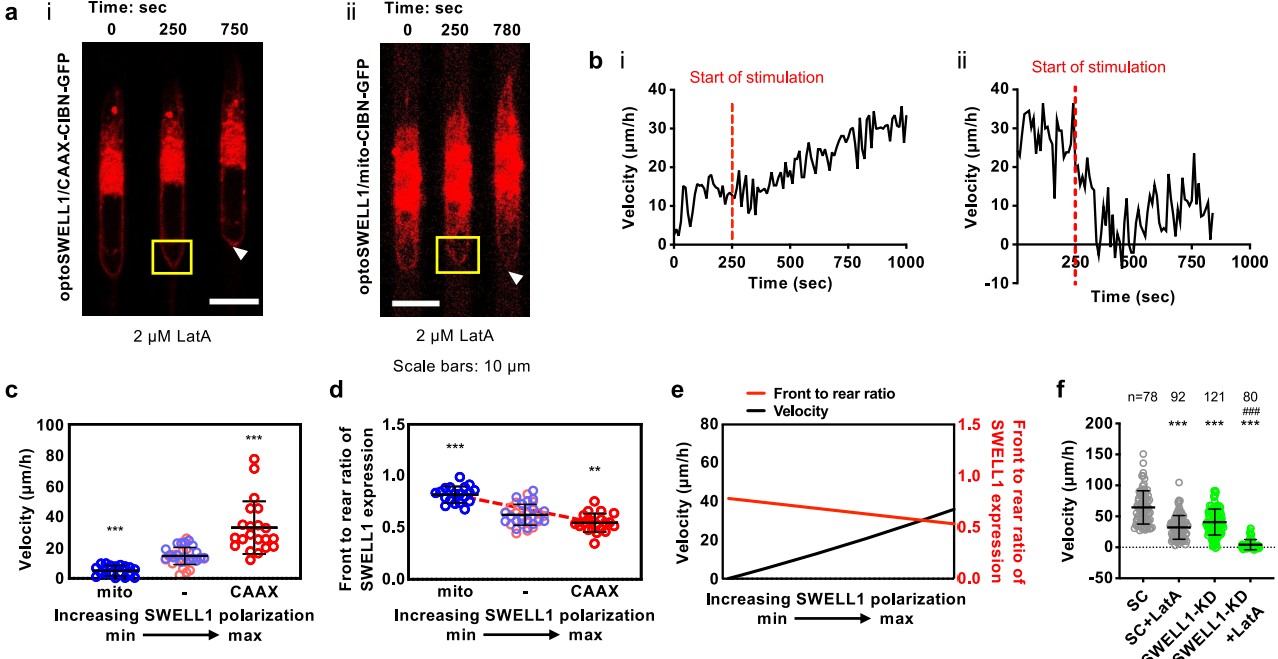

**Fig. 3 | SWELL1 polarization regulates OEM-based, F-actin-independent migration inside stiff, confining channels. a** Representative montage of MDA-MB-231 cells expressing OptoSWELL1 and (i) CAAX-CIBN-GFP or (ii) mito-CIBN-GFP inside confining channels in the presence of 2 μM LatA. Optogenetic stimulation was initiated in a region enclosed by the yellow box at the cell rear just after t = 250 sec, resulting in either SWELL1 enrichment (i) or depletion (ii) at ≥750 sec. **b** (i, ii) Instantaneous migration velocities of cells shown in **a** (i, ii) as a function of time. **c** Migration velocity and **d** front to rear ratio of SWELL1 intensity of LatA-treated MDA-MB-231 cells before (−) and after optogenetic stimulation. Data represent mean ± SD for 20 cells from 4 independent experiments. **e** A mathematical model correctly predicts that F-actin-independent migration is controlled by the extent of SWELL1 polarization. **f** Effects of SWELL1 knockdown in the presence or absence of LatA on MDA-MB-231 cell migration velocity in confinement. Data represent mean ± SD for indicated number cells from 3 independent experiments. **p < 0.01 and ***p < 0.001 relative to **c**, **d** before light stimulation or **f** SC. ###p < 0.001 relative to SC with LatA treatment or SWELL1-KD cells. Significance was determined using **c, f** Kruskal–Wallis followed by Dunn's multiple comparisons test or **d** one-way ANOVA followed by Tukey's post hoc test.

To further establish the key role of SWELL1 in regulating confined migration, we utilized the Light-induced Protein Degradation (LiPD) system to locally deplete SWELL1 expression at the cell rear. This system relies on the light-induced Cry2-CIBN heterodimer formation, which brings together the GFP binding nanobody (Cry2-Nb) and the E3 ubiquitin ligase domain (CIBN-E3) to mark GFP-tagged proteins (SWELL1-GFP) for degradation. Light stimulation at the cell trailing edge induces local depletion of SWELL1-GFP (Fig. 2e−f and Supplementary Fig. 5c), which is accompanied by reduced migration velocity in confinement (Fig. 2g and Supplementary Fig. 5d). As a control, light stimulation of SWELL1-GFP-expressing cells lacking the Cry2-Nb and CIBN-E3 constructs does not reduce SWELL1 expression or cell migration velocity (Supplementary Fig. 5c, e−g). To further validate these findings, we used optoSWELL1, and its GFP-labeled dimerization partner CIBN engineered to target SWELL1 to the mitochondrial membrane (mito-CIBN-GFP). Light stimulation at the cell rear results in progressive downregulation of SWELL1 expression locally and concomitant reduction of migration velocity (Supplementary Fig. 5h−j). Using the Mito Tracker Deep Red, we confirmed that optogenetic downregulation using the optoSWELL1 and mito-CIBN-GFP system does not interfere with the mitochondrial function of cells (Supplementary Fig. 5k−m).

To illustrate the functional contribution of OEM to confined cell migration, we optogenetically upregulated or downregulated SWELL1-GFP expression at the trailing edge of MDA-MB-231 cells inside narrow microchannels following treatment with latrunculin A (LatA, 2 μM), which abrogates actin polymerization[7]. Optogenetic enrichment of SWELL1 polarization at the rear of LatA-treated cells increases cell migration velocity in confinement (Fig. 3a−d). In contrast, optogenetic downregulation of SWELL1 expression at the

cell rear nearly halts motility (Fig. 3a−d). These experimental observations are corroborated by mathematical modeling predictions, which reveal that increasing the permeability coefficient ratio of $Cl^-$ at the back relative to the front of the cell ($\alpha^b_{Cl,p}/\alpha^f_{Cl,p}$), as a proxy of SWELL1 polarization at the trailing edge, enhances confined migration in the absence of actin polymerization (Fig. 3e). On the other hand, a decrease in $\alpha^b_{Cl,p}/\alpha^f_{Cl,p}$ ratio stalls motility (Fig. 3e), To further substantiate this experimental and theoretical finding, we further demonstrate that LatA blocks the migration of SWELL1-KD cells in confinement (Fig. 3f). Taken together, these data illustrate the cooperative roles of actin cytoskeleton and OEM in driving efficient cell migration. SWELL1 polarization at the cell trailing edge is sufficient to drive OEM-based, F-actin-independent migration in confinement. Importantly, SWELL1 regulates both the direction and efficiency of confined cell migration.

## RhoA activity regulates SWELL1 localization, whereas Cdc42 facilitates the reversal of migration direction by controlling NHE1 repolarization

Confinement induces a mesenchymal (protrusive) to amoeboid (blebbing) phenotypic switch[19,20]. Blebbing cells migrating in confinement[10,21,22] display a pill-like morphology and bear membrane blebs, which are identified as sphere-like bulges localized at the cell poles (Supplementary Fig. 6a). Because blebbing requires RhoA activation[23] (Supplementary Fig. 6a, b) and volume sensitive chloride channels are modulated by RhoA[24,25], we examined how optogenetic regulation of RhoA activity alters the spatial localization of SWELL1 using SWELL1-iRFP-expressing MDA-MB-231 breast cancer cells. MDA-MB-231 cells migrating on 2D surfaces or inside unconfined microchannels (Width = 10 μm, Height = 10 μm) exhibit a protrusive morphology (Fig. 4a). Light-induced upregulation of RhoA activity via

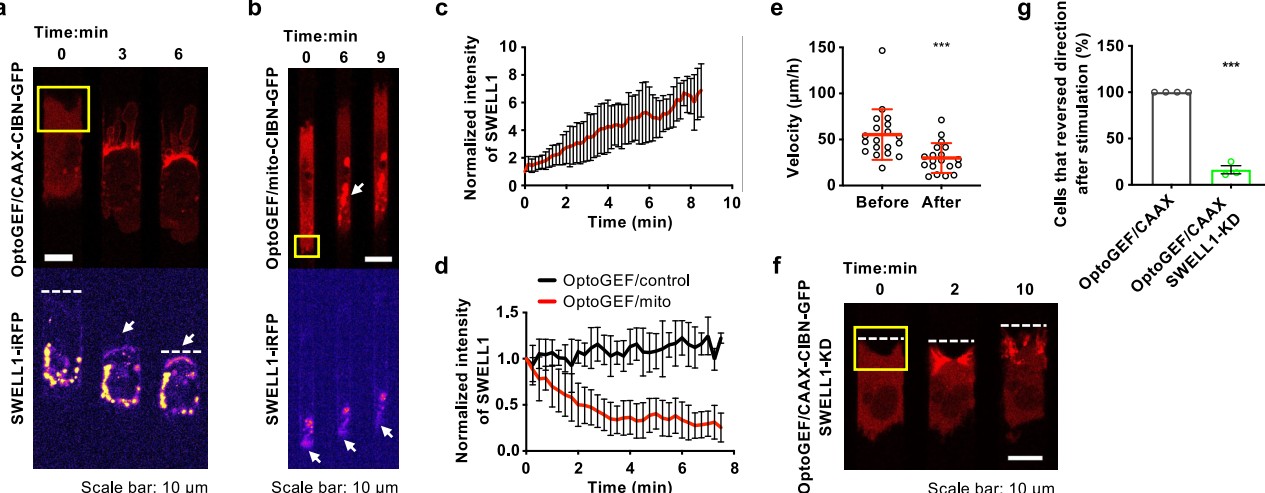

**Fig. 4 | RhoA activity regulates SWELL1 polarization. a, b** Time-lapse montage of MDA-MB-231 cells expressing SWELL1-iRFP, OptoGEF-RhoA, and **a** CAAX-CIBN-GFP or **b** mito-CIBN-GFP inside **a** 10 μm- or **b** 3 μm-wide channels. Light stimulation was applied in a region enclosed by yellow boxes just after t = 0. **a** OptoGEF-RhoA enrichment at the cell front is accompanied by local SWELL1 accumulation (white arrow), which causes migration direction reversal. The position of the cell's "old" leading edge is depicted by white dash lines in **a**. **b** Optogenetic downregulation of RhoA activity at the cell rear is accompanied by Rho-GEF accumulation in mitochondria (white arrow, top), and a reduction of SWELL1 intensity at the cell rear (white arrow, bottom). **c** Normalized SWELL1-iRFP intensity at the "old" cell front (or "new" cell rear) after optogenetically stimulating cells with optoGEF-RhoA and CAAX-CIBN-GFP. Data represent the mean ± SD for *n* = 23 cells from 5 independent experiments. **d** Normalized SWELL1-iRFP intensity at cell rear after stimulating optoGEF-RhoA in the presence (red) or absence (black; control) of mito-CIBN-GFP. Data represent the mean ± SD for *n* = 25 cells (red) and *n* = 9 cells (black) from 6 independent experiments. **e** Migration velocity of MDA-MB-231 cells expressing OptoGEF-RhoA, mito-CIBNGFP and SWELL1-iRFP before and after light stimulation. Data represent the mean ± SD for *n* = 19 cells from 6 independent experiments. **f** Time-lapse montage of a SWELL1-KD MDA-MB-231 cell expressing OptoGEF-RhoA and CAAX-CIBN-GFP. The cell's leading edge during migration inside 10 μm-wide channels is indicated by a dashed white line at the indicated times. Light-induced upregulation of RhoA activity occurred in a region enclosed by the yellow box just after t = 0 min. SWELL1-KD cell fails to reverse its migration direction following RhoA activation at its leading edge. **g** Effect of SWELL1-KD on the percentage of cells that reverse migration direction after optogenetic RhoA activation at cell leading edge. Data represent the mean ± SEM from 4 (control) or 3 (SWELL1-KD) independent experiments with *n* = 28 (control) or *n* = 25 cells (SWELL1-KD) analyzed in total. ***p < 0.001 relative to **e** before stimulation or **g** control data assessed by **e** paired or **g** unpaired two-tailed *t*-test.

optoGEF-RhoA/CAAX-CIBN-GFP[17] at the leading edge of cells inside unconfined channels enriched SWELL1 localization (Fig. 4a, c) and promoted a mesenchymal/protrusive to blebbing phenotypic switch followed by a reversal of migration direction (Fig. 4a; Supplementary Movie 6). On the other hand, light-induced downregulation of RhoA activity via optoGEF-RhoA/mito-CIBN-GFP at the trailing edge of migrating cells inside confining (3 × 10 μm²) channels locally suppressed SWELL1 localization and migration velocity (Fig. 4b, d, e). As a control, light stimulation of SWELL1-iRFP- and OptoGEF-expressing cells lacking mito-CIBN-GFP failed to alter both SWELL1 expression (Fig. 4d) and cell migration velocity (Supplementary Fig. 6c). Taken together, these data suggest that RhoA activity regulates the spatial localization of SWELL1 and modulates cell migration direction and efficiency. Remarkably, optogenetic stimulation of RhoA activity at the leading edge of SWELL1-depleted MDA-MB-231 cells migrating inside 10 × 10 μm² microchannels causes a transient retraction of the cell front but fails to reverse their migration direction (Fig. 4f, g; Supplementary Movie 7), further illustrating the critical role of SWELL1 in controlling migration direction.

Persistent cell migration requires the spatial polarization of distinct proteins at the cell front and rear, whereas disruption of cell front-to-rear polarity alters migration direction[26,27]. Because enrichment of SWELL1 expression at the cell rear confers migration directionality, and in light of the distinct polarization patterns of NHE1 and SWELL1 along the cell surface and their coordinated actions in confined cell migration, we hypothesized that efficient reversal of migration direction also requires NHE1 repolarization to the new leading edge. To test this hypothesis, cells migrating inside confining channels were subjected to a hypotonic shock (165 mOsm/l) at the cell leading edge, which caused the reversal of migration direction in 50% of the cell population (Fig. 5a, b), consistent with prior

work[7]. Importantly, immunofluorescence analysis reveals that NHE1 repolarizes to the new leading edge of cells that reversed migration direction (Fig. 5c, d). In line with the finding that only half of the cells reversed migration direction (Fig. 5a, b), the front to rear ratio of NHE1 fluorescence intensity averaged to the value of 1 (Fig. 5d), with half of the cells displaying polarization either at the old or new leading edge.

We next aimed to delineate the underlying mechanism of NHE1 repolarization to the new leading edge following the reversal of migration direction. Cdc42 is a key cell polarity protein that is typically active at the leading edge of migrating cells[26,27]. Inhibition of Cdc42 using ML141 did not alter migration velocity in confinement under isotonic conditions (Fig. 5a). However, this pharmacological intervention markedly reduced the fraction of MDA-MB-231 cells that reversed migration direction in response to hypotonic shock (Fig. 5b), which is attributed to the fact that Cdc42-inhibited cells failed to repolarize NHE1 under these conditions (Fig. 5c, d). To establish the critical role of Cdc42 activity in an efficient reversal of migration direction, we tested how optogenetic enrichment of SWELL1 expression at the cell leading edge impacts this process in the presence and absence of ML141. Cdc42 inhibition relative to vehicle control did not alter SWELL1 polarization pre- or post-optogenetic stimulation (Fig. 5e–g). Although ML141 had no effect on cell motility prior to optogenetic stimulation and did not interfere with the reversal of migration direction following light-induced upregulation of SWELL1 at the old leading edge, it markedly suppressed the migration velocity of cells after they reversed direction (Fig. 5f, g). This effect is attributed to the lack of NHE1 repolarization in Cdc42-inhibited cells. Of note, pharmacological inhibition of NHE1 via EIPA reduces migration velocity pre- and post-optogenetic enrichment of SWELL1 at the old leading edge (Supplementary

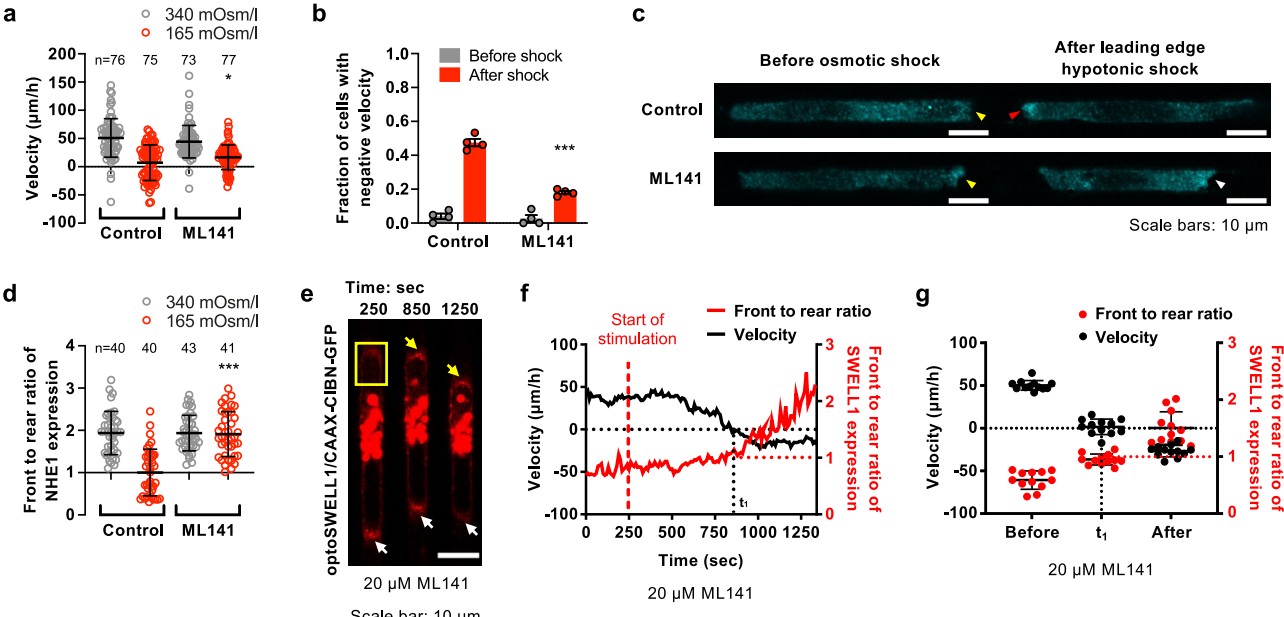

**Fig. 5 | Cdc42 facilitates the reversal of migration direction by controlling NHE1 repolarization. a** Migration velocity and **b** fraction of vehicle control and ML141-treated MDA-MB-231 cells displaying negative velocity before (gray) and after (red) hypotonic shock (165 mOsm/l) at the leading edge in confinement. Data represent the **a** mean ± SD for the indicated number of cells or **b** mean ± SEM from 3 (**a**) or 4 (**b**) independent experiments. **c** Representative images and **d** quantification of NHE1 polarization in vehicle control and ML141-treated MDA-MB-231 cells inside confining channels before (340 mOsm/l) and after (165 mOsm/l) application of hypotonic shock at the leading edge. NHE1 is polarized at the cell leading edge in control and ML141-treated cells before osmotic shock (yellow arrowheads) and repolarizes to the "new" leading edge in response to osmotic shock only in control (red arrowhead), but not ML141-treated (white arrowhead), cells. Data represent the mean ± SD from three independent experiments. **e** Time-lapse montage of an MDA-MB-231 cell expressing OptoSWELL1 and CAAX-CIBN-GFP treated with ML141. Optogenetic SWELL1 enrichment at the cell leading edge was initiated in a region enclosed by the yellow box just after t = 250 sec. Gradual SWELL1 accumulation at the cell front (yellow arrowhead) is accompanied by SWELL1 reduction at the cell rear (white arrowhead). At t = $t_1$ = 850 sec, SWELL1 is equally distributed at both cell poles. **f** Migration velocity (black line) and front to rear ratio of SWELL1 intensity (red line) of the cell shown in **e**. **g** Migration velocity (black dots) and front to rear ratio of SWELL1 intensity (red) of MDA-MB-231 cells expressing OptoSWELL1 and CAAX-CIBN-GFP that were treated with ML141 before stimulation, at t = $t_1$ and t ≥ 1,200 sec after stimulation. Data represent the mean ± SD for 12 cells from 3 independent experiments. *$p < 0.05$ and ***$p < 0.001$ relative to control data after shock assessed by two-tailed unpaired $t$-test.

Fig. 6d, e). Cumulatively, these data illustrate that Cdc42 activity is required for efficient cell reversal.

## Dual NHE1 and SWELL1 knockdown blocks breast cancer cell extravasation and metastasis in vivo

In view of the critical roles of NHE1 and SWELL1 in cell dissemination from breast cancer spheroids and cell migration in 3D collagen gels and confining channels in vitro, we examined their functional contributions to breast cancer metastasis in vivo. To this end, luciferase- and GFP-labeled SC, NHE1-KD, SWELL1-KD, and dual NHE1/SWELL1-KD MDA-MB-231 breast cancer cells were subcutaneously injected into the 4th mammary fat pad of NOD-SCIDγ (NSG) mice. Bioluminescence imaging reveals that SC, single- and dual-KD cells formed tumors that grew at similar rates for up to 3 weeks (Fig. 6a, b). All mice were sacrificed at week 4 when the bioluminescence signal of the primary tumor reached saturation.

Bioluminescence imaging analysis at necropsy revealed that all mice (10/10) injected with SC cells developed metastases in the bone and brain, whereas only four out of ten (4/10) mice with dual NHE1/SWELL1-KD cells displayed metastases in these tissues (Fig. 6c). SC cells were also more efficient than dual-KD cells (10/10 versus 6/10) in generating metastases in the axillary lymph nodes (Fig. 6c). Moreover, individual NHE1 or SWELL1 knockdown reduced the frequency of metastasis in these tissues (Fig. 6c). Although all mice from the SC group and nearly all from the dual-KD exhibited metastases in the liver and lung, bioluminescence image analysis of the surgically-isolated tissues revealed a 10-fold and 20-fold decrease in the metastatic burden, respectively, for the dual-KD tumor cells (Fig. 6d, e). To independently validate the reduced metastatic burden detected in the liver and lungs of mice injected with dual-KD as opposed to SC cells, DNA was extracted from these tissues, and the amount of human DNA was analyzed using quantitative PCR (qPCR) with primers specific for human long interspersed nuclear elements(hLINE)[12,28]. Dual-KD relative to SC cells displayed a six-fold and five-fold reduced amount of human DNA in the livers and lungs, respectively (Fig. 6f). Individual NHE1 or SWELL1 knockdown suppressed metastatic burden in these tissues, as assessed by bioluminescence image analysis (Fig. 6e), whereas a trend was detected by qPCR, which reached statistical significance for the NHE1-KD cells in the lung (Fig. 6f). Taken altogether, these in vivo data are in accord with in vitro findings showing that dual relative to single KD has a significantly higher inhibitory effect on the migration of luciferase-labeled cells (Supplementary Fig. 7). Tissue samples from representative SC and KD specimens were also processed for immunohistochemistry against GFP, which was used as a specific marker for the transplanted human tumor cells, as well as hematoxylin and eosin staining (Fig. 6g). These images confirm the consistent presence of metastatic human SC cells in the livers and lungs of mice and the marked reduction of dual-KD cells.

Tumor cell extravasation is a critical step for the dissemination of cancerous cells to distant organs in the body. To examine the functional involvement of NHE1 and SWELL1 in tumor cell extravasation, we employed the avian embryo cancer cell extravasation assay[29,30], which permits real-time visualization of this process at excellent optical resolution. When injected into the embryo bloodstream, SC cells robustly extravasated from the CAM vasculature, with the majority leaving the vasculature by the 8 h timepoint (Fig. 6h). In marked

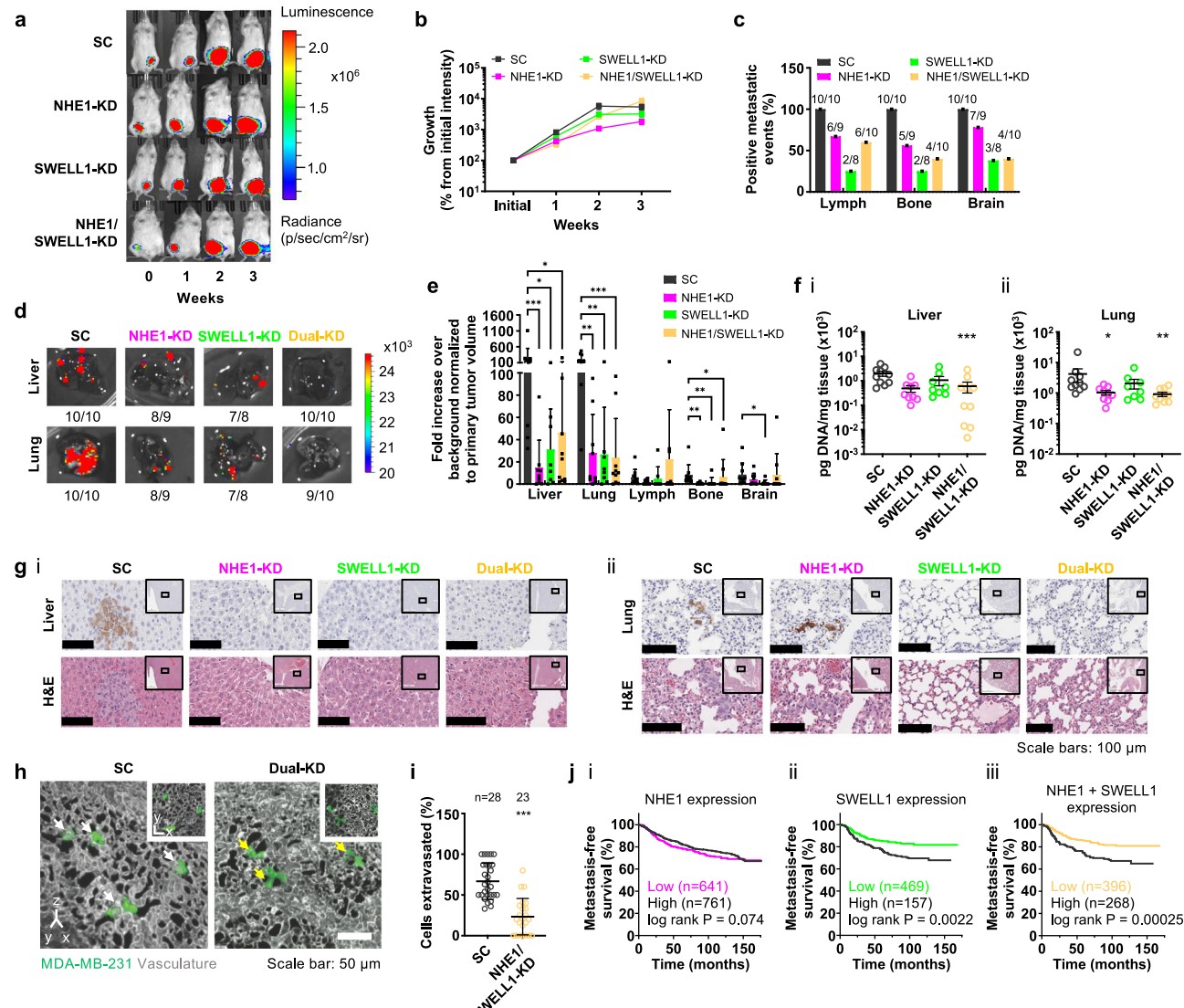

**Fig. 6 | Dual NHE1 and SWELL1 knockdown blocks breast cancer cell extravasation and metastasis. a**, **b** Bioluminescent images of mice following subcutaneous injection of SC, single NHE1-KD or SWELL1-KD, or dual-KD cells into the mammary gland. Data represent mean ± SEM from 10 mice per group at the indicated timepoints, except for SWELL1-KD ($n = 9$ at week 2; $n = 8$ at week 3). **c** Percentage of mice with positive metastatic events in the lymph node, bone and brain for the indicated number of mice. One mouse in **a**, **b** died before imaging for metastasis ($n = 9$ for NHE1-KD). **d** Bioluminescent images of the liver and lung of mice at endpoint. **e** Quantification of bioluminescent signal from liver, lung, lymph nodes, bone, and brain. Data represent mean ± SD for the same number of mice shown in **c**, except for one extreme outlier in the lymph nodes of a mouse injected with dual-KD cells. **f** Amount of human DNA in the (i) liver or (ii) lung of mice at endpoint as determined by qPCR. Data represent mean ± SEM for the same number of mice shown in **c**. **g** Representative 10× images of adjacent (i) liver or (ii) lung sections immunostained for GFP or hematoxylin and eosin (H&E). Insets show the surrounding area at 2×, and indicate the area displayed at 10×. **h** 3D reconstructions of CAM vasculature with extravasating cells engineered to express either SC or dual NHE1 and SWELL1-KD. White arrows indicate extravasated cells; yellow arrows denote cells that are still located within the vasculature. Insets show single optical planes of the same fields of view. **i** Quantification of cell extravasation. Data represent the mean ± SD for the indicated number of cells from three independent experiments. **j** Kaplan–Meier analysis of NHE1 expression (i), SWELL1 expression (ii), NHE1 and SWELL1 expression (iii) and distant metastasis-free survival for breast cancer patients. *$p < 0.05$, **$p < 0.01$, and ***$p < 0.001$ relative to SC (**e**, **f**, **i**) assessed by **e** Kruskal–Wallis followed by Dunn's test or **f** one-way ANOVA followed by Tukey's test after log transformation or **i** two-tailed unpaired Mann–Whitney test. Significance in **j** was computed using the Cox–Mantel (log rank) test as provided by KM-plotter. Cell model: MDA-MB-231.

contrast, dual NHE1/SWELL1-KD suppressed breast cancer cell extravasation more than two-fold with the majority of these cells remaining inside the CAM vascular network (Fig. 6h, i). These findings are in agreement with the data obtained in the in vitro setting and a metastatic murine model.

To provide further support for the role of NHE1 and SWELL1 in breast cancer metastasis, patient distant metastasis-free survival (DMFS) time and gene expression data were analyzed using the KM-plotter database[31], which combines datasets from GEO, EGA, and TCGA. Patients were split into two groups based on their expression levels of NHE1 and SWELL1. The cutoff producing the greatest separation of DMFS between the 2 groups is shown in Fig. 6j. Comparing the lowest and highest tertiles or quartiles produced similar results. Kaplan–Meier survival analysis shows that patients expressing high levels of both NHE1 and SWELL1 had lower DMFS (Fig. 6j (iii)), suggesting that inhibition of NHE1 and SWELL1 may represent a potential therapeutic regimen for suppressing breast cancer dissemination and metastasis in vivo.

## Discussion

Cell migration is a pivotal step in the metastatic dissemination of cancer cells from a primary tumor to distant organs in the body. Cell motility is governed by cell-matrix interactions, the actomyosin cytoskeleton, and cell volume regulation as proposed by OEM[7]. According to OEM, a cell migrating in confinement establishes a spatial gradient of ion transporters, ion channels, and AQPs in the cell membrane so that local swelling at the leading edge and shrinkage at the trailing edge, respectively, facilitate net cell movement[7]. We previously reported that NHE1, which polarizes at the cell leading edge, can support confined migration even after complete disruption of F-actin[7]. We herein show that NHE1 promotes isosmotic cell swelling consistent with its role in RVI[4]. Moreover, we determined that SWELL1 and AQP4 are preferentially enriched at the cell rear of migrating cells in confinement and mediate local cell shrinkage via RVD. The coordinated action of isosmotic swelling and shrinkage at the cell poles mediated by NHE1 and SWELL1, respectively, due to their distinct polarization patterns and roles in RVI and RVD, supports efficient confined migration. We postulate that the polarization of SWELL1 at the cell trailing edge mediates local RVD due to an outflow of Cl⁻ ions and water, which results in rear membrane shrinkage and decreased cell volume relative to its equilibrium state. This reduced cell volume is compensated by RVI, which occurs at the cell leading edge due to the local enrichment of NHE1 expression. When the cell volume exceeds its equilibrium state due to RVI, SWELL1 is reactivated, and the coordinated RVD/RVI cycle is repeated. Through this feedback loop and dynamic regulation of water/ion fluxes, cells maintain their volume and migrate efficiently in confinement. Analogous to squeezing a soft, porous material filled with water such as a sponge at one end, SWELL1-mediated outflow of Cl⁻ ions and water at the cell rear dissipate pressure towards the extracellular environment, while its rear end shrinkage concurrently propagates pressure towards the nucleus, causing its forward translocation. This coordinated RVD/RVI cycle, which involves SWELL1-dependent rear-end retraction and forward nuclear translocation coupled with NHE1-mediated leading edge protrusion, is responsible for efficient cell locomotion.

We further demonstrate that dual NHE1 and SWELL1 knockdown blocks cancer cell dissemination from breast cancer spheroids and reduces their motility in 3D as well as breast cancer cell extravasation and metastasis in vivo without affecting tumorigenesis. The inhibitory effects of combined NHE1 and SWELL1 depletion are stronger than their individual knockdowns both in vitro and in vivo, consistent with their coordinated functional roles.

Recent work has shown that SWELL1 promotes the motility of various cancer cell types, such as hepatocellular carcinoma (HCC)[32] and gastric cancer cells[33], as evidenced by wound healing and transwell migration assays. Yet, how SWELL1 impacts cell motility remains elusive. Using an integrated experimental and mathematical approach, we demonstrate that SWELL1 polarization at the cell rear mediates not only shrinkage of the cell trailing edge in accord with its role in RVD, but, most importantly, also confers migration direction. Optogenetic enrichment of SWELL1 at the cell leading edge reverses migration direction, whereas the equal distribution of SWELL1 at the cell poles ceases motility. Using optogenetic tools, we further demonstrate that SWELL1 localization is regulated by RhoA activity. As such, upregulation of RhoA activity at the cell front promotes local SWELL1 enrichment, thereby reversing migration direction. Importantly, the inability of SWELL1-knockdown cells to reverse migration direction in response to optogenetic RhoA stimulation at the cell front illustrates the indispensable role of SWELL1 in this process. It is noteworthy that efficient cell reversal also requires Cdc42, which controls NHE1 repolarization to the "new" leading edge. Importantly, these data reveal the crosstalk between ion transporters/channels and key constituents of the cell cytoskeleton (RhoA and Cdc42) in the process of cell migration.

Despite general consensus in the literature regarding the critical role of SWELL1 in cell migration, it has also been reported that SWELL1 is dispensable for the migration of HCT116 colon carcinoma cells and U251 and U87 glioblastoma cells[34]. Cell phenotypic differences might provide a potential explanation for this discrepancy. Because RhoA activation induces SWELL1 enrichment as well as a blebbing phenotype[10,11], whereas RhoA inhibition depletes SWELL1 localization and is accompanied by a protrusive/mesenchymal phenotype[11], we postulate that SWELL1 facilitates the migration of cells displaying a blebbing phenotype.

There are conflicting data regarding the potential role of SWELL1 in cell proliferation. For instance, SWELL1 overexpression has been reported to induce the proliferation of HCC cells, whereas its depletion has opposite effects[32]. Along these lines, SWELL1 knockdown suppresses both primary tumor growth and metastasis of HCC cells in vivo[32]. However, recent findings have linked cell survival to SWELL1 activity and/or expression, albeit only under hypertonic conditions[35]. Moreover, others have shown that SWELL1 does not affect cell proliferation[34], which is in line with our in vitro findings and the lack of any effect of SWELL1 depletion on primary tumor growth. Although SWELL1 knockdown tended to decrease the metastatic burden in the liver and lungs of mice as assessed by qPCR and bioluminescence (in the case of the liver), metastasis in these organs was consistently and markedly inhibited upon dual NHE1 and SWELL1 knockdown. This dual intervention also blocked breast cancer cell extravasation in the CAM model. The in vivo efficacy of dual NHE1 and SWELL1 depletion, which correlates with our in vitro findings using diverse complementary assays and the Kaplan–Meier survival analysis, establishes the physiological relevance of OEM.

## Methods

### Experimental methods

All mouse experiments were performed in accordance with the Institutional Animal Care and Use Committee procedures and guidelines of the University of Maryland at Baltimore under approved protocol number 0219006. All procedures involving chick embryos were approved by the University of Alberta Institutional Animal Care and Use Committee (IACUC).

### Cell culture

Human MDA-MB-231 cells[12] (ATCC, catalog number: HTB-26) were cultured in DMEM (Gibco) supplemented with 10% heat-inactivated FBS (Gibco) and 1% penicillin/streptomycin (10,000 U/mL, Gibco). SUM159 cells[12] were provided by Denis Wirtz (Johns Hopkins University) and were grown in Ham's F-12 medium (Corning Cellgro) plus 5% FBS, 1% penicillin/streptomycin, 1 μg/mL hydrocortisone (Sigma–Aldrich) and 5 μg/mL insulin (Sigma–Aldrich). PTEN-/-/KRAS(G12V) MCF-10A cells were a gift from Michele I. Vitolo (University of Maryland at Baltimore) and were cultured as described previously[13]. MDA-MB-231-luciferase cells were created and cultured as described previously[12]. Cells were maintained in an incubator at 37°C with 95% air/5% CO₂ and passaged upon 60–80% confluency every 3–5 days. Cells were routinely checked for mycoplasma contamination via PCR using the primers: F-(5′-GGGAGCAAACAGGATTAGATACCCT-3′) and R-(5′-TGCACCATCTGTCACTCTGTTAACCTC-3′).

### Cloning, lentivirus production, and cell transduction

To generate plasmids with shRNA lentiviral vectors, we subcloned the targeting sequences or nontargeting scramble control into the pLVTHM lentiviral plasmid (Addgene, plasmid 12247, a gift from Dider Trono) using MluI and ClaI as restriction sites. The target sequences are as follows:

nontargeting scramble control sh1 (5′-GCACTACCAGAGCTAAC TCAGATAGTACT-3′),

human sh1NHE1 (5′-GACAAGCTCAACCGGTTTAAT-3′),

human sh2NHE1 (5′-CCAATCTTAGTTTCTAACCAA-3′).

In addition, we subcloned the targeting sequences or nontargeting scramble control into the pLKO.1 lentiviral plasmid (Addgene, plasmid 8453, a gift from B. Weinberg) using AgeI and EcoRI as restriction sites. The target sequences are:

nontargeting scramble control sh1, human NHE1 sh1, and human NHE1 sh2, as shown above,

human sh1SWELL1 (5′-GGTACAACCACATCGCCTA-3′),
human sh2SWELL1 (5′-GAGCAAGTCTCAAGAGCGC-3′),
human shAQP4 (5′-CCAAGTCCGTCTTCTACAT-3′). Sequence integrity and orientation were verified by Sanger Sequencing (JHU Genetic Resources Core Facility).

The pLVTHM, pLKO.1, pLenti.PGK.LifeAct-GFP.W (plasmid 51010, a gift from Rusty Lansford), pLenti.PGK.H2B-mCherry (plasmid 51007, a gift from Rusty Lansford), psPAX2 (plasmid 12260, gift from Didier Trono), and pMD2.G (plasmid 12259, a gift from Didier Trono) plasmids were purchased from Addgene.

For lentivirus production, 293 T/17 cells were cotransfected with psPAX2, pMD2.G, and the lentiviral plasmid of interest. The media was refreshed after 24 h. Lentivirus was harvested 48 h after transfection, filtered through 0.45 μm filters (Fisher Scientific), and purified by centrifugation (50,000 × $g$ for 2 h at 4 °C). Next, cells were transduced for 48 h with a medium containing lentiviral particles. Puromycin (0.5 μg/mL, Gibco) was added to the cell culture media 48 h after transduction, and this concentration was maintained to select cells transduced with pLKO.1 vectors. In all in vitro and in vivo experiments involving SC, single- and dual-KD cells, proper controls were included by transducing cells with the corresponding nontargeting sequences in the appropriate vectors (Supplementary Table 1).

## Plasmid transfection

The SWELL1-GFP plasmid was a kind gift from Thomas J. Jentsch (Leibniz-Institut für Molekulare Pharmakologie (FMP), Berlin). The AQP4-mCherry plasmid was provided by Antonio Frigeri (University of Bari). The plasmids for the light-induced protein degradation system, (LiPD) pRing-CIBN-IR and pGBP-PHR-IR, were kind gifts from Heinrich Leonhardt (University of Munich). MDA-MB-231 cells, at 50–60% confluency, were transiently transfected with Lipofectamine 3000 reagent following the manufacturer's recommendation. Only in the case of SWELL1-GFP, a stable cell line was generated following treatment with G418 (Corning) and sorting.

## Generation of plasmids for optogenetic experiments

SWELL1-iRFP was created by replacing GFP from the SWELL1-GFP plasmid with iRFP. iRFP was amplified using piRFP670-N1 (Addgene, plasmid 45457). SWELL1-iRFP was inserted into lentiviral backbone pLV-EF1a-IRES-Puro (Addgene, plasmid 85132). Forty-eight hours post-shRNA transduction, puromycin (0.5 μg/mL, Gibco) was added to a fresh cell culture medium to select stably transduced cells expressing SWELL1-iRFP.

ARHGEF11(DHPH)-Cry2-mCherry (OptoGEF), CAAX-CIBN-GFP, and mito-CIBN-GFP were gifts from Dr. Xavier Trepat (Institute for Bioengineering of Catalonia). Cry2-mCherry was amplified from ARHGEF11(DHPH)-Cry2-mCherry and inserted into pLV-EF1a-IRES-Hygro (Addgene, plasmid 85134). SWELL1 was amplified from SWELL1-iRFP, and then inserted into pLV-EF1a-IRES-Hygro-Cry2-mCherry to create SWELL1-Cry2-mCherry (OptoSWELL1). Hygromycin B (500 μg/mL, ThermoFisher Scientific) was added to a fresh medium 48 h post-transduction to select cells stably transduced with OptoSWELL1.

## Microfluidic device fabrication, cell seeding, cell treatment, live-cell imaging, and analysis

PDMS-based microfluidic devices containing an array of parallel microchannels of prescribed height (10 μm), width (3 μm), and length (200 μm) were fabricated as described previously[36–38]. The microchannel dimensions were verified by a laser profilometer. Microchannels were sandwiched orthogonally by 2D-like seeding and media channels[36–38]. Prior to migration assays, assembled microfluidic devices were incubated with rat tail collagen I (20 μg/ml, Thermo Fisher Scientific) for at least 1 h at 37 °C in the presence of 95% air/5% CO₂. Migration experiments were performed in DMEM containing 10% heat-inactivated FBS (Gibco) and 1% penicillin/streptomycin (10,000U/ml, Gibco). No chemotactic stimulus was applied in these experiments. Twenty microliters of cell suspension (4 × 10⁶ cells/ml) in serum-containing medium were added to the device seeding inlet. In select experiments, cells were treated with the following pharmacological agents or corresponding vehicle controls: 5-(N-Ethyl-N-isopropyl) amiloride (EIPA, 40 μM, Sigma–Aldrich), DCPIB (40 μM, Tocris Biosciences), bumetanide (30 μM, Santa Cruz Biotechnology), Lat-A (2 μM, Sigma–Aldrich), ML141 (10 μM, Santa Cruz Biotechnology). In these assays, a medium containing either the drug or the vehicle control was added to all inlet and outlet wells of the device at the onset of the migration experiment, unless otherwise stated. In Lat-A assays, Lat-A-containing medium was added only after the cells had fully entered the microchannels.

Time-lapse images were recorded in 10 min intervals for up to 20 h in an inverted Nikon Eclipse Ti microscope (Nikon, Tokyo, Japan) equipped with a stage-top incubator (Okolab, Pozzuoli, Italy, or Tokai Hit, Shizuoka, Japan) at 37 °C and 95% air/5% CO₂, automated controls (NIS-Elements, v. 4.13.05; Nikon) and a ×10/0.30 numerical aperture Ph1 objective. Cell migration analysis was performed as previously described[10,39]. Briefly, live-cell videos were exported to ImageJ (v.2.0.0/1.51 h; National Institute of Health, Bethesda, Maryland). The tracks of individual cells that had fully entered the microchannels were obtained manually via Manual Tracking (Cordelières F, Institut Curie, Orsay, France) plugin. Cell migration velocity was calculated using a custom MATLAB script (MathWorks, Natick MA). Cell entry time was defined as the time interval from the point that a cell's leading edge initiated entry into the microchannel until its trailing edge had fully entered the microchannel, and was calculated manually. The cell longitudinal area was measured by manually outlining the cell periphery in ImageJ.

## Microfluidic assays in Na⁺ free or Cl⁻ low media

To test the effect of extracellular Na⁺ on cell motility, select microfluidic assays were performed with a self-assembled NaCl-containing medium (control) and N-Methyl-D-glucamine-chloride (NMDG-Cl⁻)-containing medium (Na⁺ free). NaCl-containing medium was composed of 140 mM NaCl, 2.5 mM KCl, 0.5 mM MgCl₂, 1.2 mM CaCl₂, 5 mM glucose and 10 mM HEPES, with pH and osmolarity adjusted to 7.4 and 300–305 mOsm/L, respectively. To generate the Na⁺ free medium, NaCl was substituted with NMDG-Cl⁻. Cells were seeded and then incubated in a serum-containing medium for at least 3 h at 37 °C in the presence of 95% air/5% CO₂. After cells fully entered the microchannels, the medium was removed, and a control medium or Na⁺ free medium was added to all inlet and outlet wells of the device. Time-lapse images were recorded every 10 min for up to 6 h in 100% air, with all other imaging settings remaining the same as the regular migration assay.

To test the effect of extracellular Cl⁻ on cell motility, select assays were performed with high glucose (4.5 g/L) DMEM (regular medium), self-assembled NaCl-containing medium (control), and Na⁺-glutamate-containing medium (Cl⁻ low). The control medium was prepared by adding all the components present in the formulation of commercial DMEM with pH and osmolarity adjusted to 7.4 and 340 mOsm/L, respectively. To generate Cl⁻ low medium, NaCl was substituted with a corresponding amount of Na⁺-glutamate. Regular, control, or Cl⁻ low medium was added to all inlet and outlet wells of the microchannel device after the cells were seeded. Time-lapse images were recorded in 10 min intervals for up to 6 h at 37 °C in the presence of 95% air/5% CO₂, with all other image settings remaining the same as the regular migration assay.

## Spheroid formation, and 3D collagen invasion assay

Spheroids were formed as previously described[40]. Briefly, growth factor reduced Matrigel was diluted with DMEM containing 10% heat-inactivated FBS and 1% penicillin/streptomycin at 1:3 ratio. Fifty microliters of the diluted Matrigel were transferred to a 96-well plate (Falcon) and polymerized for 1 h at 37 °C in a cell culture incubator, while the rest of the diluted Matrigel was kept on ice. $2 \times 10^3$ breast cancer cells were suspended in 50 μL ice-cold Matrigel and gently plated in different wells pre-coated with polymerized Matrigel followed by incubation at 37 °C and 5% CO$_2$ in a cell culture incubator. ~1.5 h later, 100 μL of prewarmed DMEM containing 10% FBS and 1% P/S was added in each well. The cell culture medium was replaced every two days, and spheroids were used in invasion assays ~10–15 days later.

3D collagen invasion assays using spheroids were performed as previously described[41]. Briefly, 3 mL of rat tail collagen type I (Corning) was gently mixed with 375 μL of 10× DMEM−low glucose (Sigma). The pH of the mixture was slowly adjusted to physiological levels with NaOH. After 1 h incubation on ice, 25 μL of the mixture were added to a 24-well plate (Falcon) and incubated at 37 °C for 1 h. Spheroids were collected into 1.5 ml Eppendorf tubes by gently disrupting the Matrigel with ice-cold DMEM. The Eppendorf tube was incubated in ice to further depolymerize the Matrigel for >10 min. Spheroids were isolated by $2665 \times g$ centrifugations for 5 min, and resuspended into 100 μL of the collagen mixture. Next, 100 μL of the spheroid-collagen mixture were plated in each well and incubated at 37 °C for 1–1.5 h. After collagen polymerization, 500 μL prewarmed cell culture media was added to each well.

Time-lapse images were recorded in 20 min intervals for ~30 h in an inverted Nikon Eclipse Ti microscope (Nikon) equipped with a stage-top incubator (Okolab or Tokai Hit) at 37 °C and 95% air/5% CO$_2$, automated controls (NIS-Elements, Nikon) and a ×10/0.30 numerical aperture Ph1 objective. First-cell dissociation times were obtained using NIS-Elements (Nikon) by manually measuring the time required for the first cell to fully detach from the spheroid. Cell velocity, mean squared displacement (MSD), and cell trajectory were calculated using a custom-made MATLAB script. Normalized area expansion and circularity were determined using ImageJ by outlining the spheroid at $t = 0$ and 12 h using polygonal regions of interest. In select experiments, spheroids were placed on 2D collagen I (20 μg/ml)-coated surfaces, and cell dissociation was tracked in real-time (Supplementary Movie 4).

To investigate the spatiotemporal distribution of SWELL1 during dissociation of SWELL1-GFP-tagged cells from 3D spheroids embedded in collagen gels, time-lapse confocal images were recorded at 10 min intervals for up to 30 h using a Nikon A1 confocal microscope equipped with a ×20 air objective, a 488 nm laser and NIS-Elements software (v. 5.02.01).

## Cell volume measurements

Lifeact-GFP-labeled cells were imaged using a Nikon A1 confocal microscope with a ×60 oil objective and a 488 nm laser. Cells were visualized with Imaris (v. 9.7.0; Bitplane, Zurich, Switzerland), and their volume was measured from confocal Z-stacks with a step of 0.5 μm using a custom MATLAB (v. R2016b; MathWorks, Natick, MA) script, as described previously[10].

## Patch-clamp experiments

Whole-cell recordings were obtained as previously described[42] using an Axon 200 A amplifier (Axon Instruments, San Jose, CA). Currents were acquired at 33 kHz and filtered at 1 kHz. The pClamp8 software (v. 10; Axon Instruments) was used for pulse generation, data acquisition, and subsequent analysis. LRRC8A-like chloride currents were measured in cells clamped at 0 mV and pulsed for 400 ms from −100 mV to +100 mV in 50 mV steps every 30 sec. $I_{Cl}^-$ whole-cell currents were measured using pipettes (2-3 MΩ) filled with a solution containing 100

mM N-methyl-D-glucamine chloride (NMDGCl$^-$), 1.2 mM MgCl$_2$, 1 mM EGTA, 10 mM HEPES, 2 mM Na$_2$ATP, and 0.5 mM Na$_3$GTP (pH 7.3 and 300 mOsm/l). The external solution contained NMDGCl$^-$ at 100 mM (for iso and hypotonic conditions) or 185 mM (hypertonic conditions), 0.5 mM MgCl$_2$, 5 mM KCl, 1.8 mM CaCl$_2$, 5 mM glucose, and 10 mM HEPES, pH 7.4. Osmolarity was adjusted to 310 (isotonic), 220 (hypotonic), with mannitol.

## Optogenetic control of RhoA activity and SWELL1 localization

Optogenetic tools were utilized to control the subcellular activation of RhoA with high spatiotemporal accuracy, using the cryptochrome 2 (Cry2)-CIBN light-gated dimerizer system[17,18]. This system relies on the fusion of the catalytic (DHPH) domain of the RhoA-GEF, ARH-GEF11, to Cry2-mCherry (optoGEF-RhoA) and its GFP-labeled dimerization partner, CIBN, engineered to bind to the plasma via the CAAX anchor (CAAX-CIBN-GFP) or mitochondrial membrane via mito-CIBN-GFP. MDA-MB-231 cells, stably transduced with either CAAX-CIBN-GFP or mito-CIBN-GFP, ARHGEF11(DHPH)-Cry2-mCherry and SWELL1-iRFP, were used to assess the effect of spatiotemporal alterations of RhoA activity on SWELL1 localization. To this end, cells migrating inside confining channels were monitored in real-time by imaging the mCherry channel to identify the leading and trailing edges. Light stimulation was performed with a 488 nm laser at 1% power for 1 sec on a rectangular area placed either at the cell leading or trailing edge. Stimulations were repeated at 10 sec intervals for 10–30 min to enable consistent localization of ARHGEF11 to the membrane or mitochondria. mCherry and iRFP670 images were recorded after each stimulation to monitor the localization of ARH-GEF11 and SWELL1.

To directly establish the role of SWELL1 polarization in the direction and efficiency of migration, MDA-MB-231 cells, stably transduced with either CAAX-CIBN-GFP or mito-CIBN-GFP, and OptoSWELL1, were subjected to light stimulation as described above. Briefly, cell migration and SWELL1 localization were monitored by imaging the mCherry channel in real-time. Cell velocity and front-to-rear ratio of SWELL1 expression were quantified using a custom MATLAB script.

In select optogenetic experiments, a medium containing either a pharmacological agent or its vehicle control was added to all inlet and outlet wells of the device only after the cells had fully entered the microchannels. Confocal imaging was initiated at least 30 min after the addition of the drug-containing medium.

## Light-induced protein degradation (LiPD) assays

LiPD assays were performed as recently described[43]. MDA-MB-231 cells expressing SWELL1-GFP were transfected with pRing-CIBN-IR and pGBP-PHR-IR plasmids (see above). Cells were seeded in microfluidic devices and imaged using a Nikon A1 confocal microscope. A 561 nm laser was used to detect the co-expressed DsRed signal and monitor cells in real-time. Light stimulation was performed with a 488 nm laser at 5% laser power for 1 sec on a rectangular area marked at the cell trailing edge. Stimulations were repeated at 10 sec intervals for up to 30 min. GFP images were recorded after 20 rounds of stimulation to monitor the extent of SWELL1 expression.

## Mitochondria function assay

The assay was performed using the MitoTracker® Deep Red FM (M22426, ThermoFisher Scientific). Briefly, MitoTracker® Deep Red FM working solution was prepared at 1 mM according to the manufacturer's instructions, and then diluted to 25 nM using a cell medium. MitoTracker® Deep Red FM (25 nM) and Hoechst 33342 (ThermoFisher Scientific, H3570) at 1:5000 dilution were added to all inlet and outlet wells of microfluidic devices after the cells had fully entered the microchannels. Devices were then placed for 45 min in an incubator at 37 °C with 95% air/5% CO$_2$. Next, the liquid was removed from

microfluidic devices, and replaced with the fresh prewarmed medium. Cells were imaged using a Nikon A1 confocal microscope, and mitochondria intensity was measured using ImageJ. Normalized mitochondria intensity was calculated relative to DAPI intensity of the nucleus.

### Fluorescence lifetime imaging microscopy (FLIM) of RhoA FRET sensors

Confocal FLIM of live MDA-MB-231 cells stably expressing the RhoA2G sensor was carried out as outlined in refs. [10,22], using ZEN 2.3 SP1 FP3 (black; Zeiss, Jena, Germany) and SymPhoTime 64 (v. 2.4; PicoQuant, Berlin, Germany).

### Hypotonic shock assays

Hypotonic solutions were prepared, and their osmolarity was measured as previously described[7,10]. After 1.5 h of live-cell imaging in drug- or vehicle control-containing serum-free isotonic medium, the medium in lower and upper wells of the microfluidic devices was replaced with serum-free isotonic or hypotonic medium, respectively, containing the drug or its matching vehicle control. In all pre- and post-shock experiments, the uppermost inlet also contained 10% heat-inactivated FBS. Phase-contrast time-lapse images were recorded at 5 min intervals for 2 h. In select experiments, cells were prepared for immunofluorescence analysis and evaluated under confocal optics.

### Immunofluorescence

For immunostaining with SWELL1 (LRRC8A) antibody (a kind gift from Thomas J. Jentsch[9]), cells were fixed in pre-cooled methanol (Fisher Chemical) at −20 °C for 10 min, followed by incubation with 30 mM glycine (Sigma) in PBS for 5 min at room temperature. Cells were incubated overnight with the primary antibody (1:100) at 4 °C, followed by washing 3× with PBS, and then 1 h incubation with a secondary antibody (1:100) in PBS containing 0.1% Triton X-100 supplemented with 3% BSA at 4 °C. For immunostaining with NHE1, AQP4, or Ki-67 antibodies, cells were fixed with 4% formaldehyde solution (ThermoFisher Scientific), permeabilized with 0.1% Triton® X-100 (Sigma–Aldrich), blocked with 1% bovine serum albumin (Sigma–Aldrich), immunostained, and imaged with an A1 confocal microscope. Primary antibodies were used at the following concentrations: anti-NHE1 (1:50; mouse, clone 54, Santa Cruz Biotechnology, sc-136239), anti-AQP4 (1:50; mouse, clone 4/18, Santa Cruz Biotechnology, sc-32739), or anti-Ki-67 (1:800; clone 8D5, Cell Signaling Technology). Following overnight incubation with primary antibodies at 4 °C, specimens were washed 3× with PBS, and then secondary antibodies (obtained from Invitrogen) were applied for 1 h at room temperature: Alexa Fluor 488 goat anti-mouse immunoglobulin-G (IgG) (H + L) (1:100), Alexa Fluor 568 goat anti-rabbit IgG (H + L) (1:200), Alexa Fluor Plus 647 goat anti-mouse IgG (H + L) (1:100), or Alexa Fluor Plus 647 goat anti-rabbit IgG (H + L) (1:100). Nuclei were also stained with Hoechst 33342 (1:2500, ThermoFisher Scientific, H3570).

### Quantification of NHE1 and SWELL1 polarization

A custom MATLAB script was used to segment NHE1 or SWELL1 intensity at the cell front and rear, and exclude the signal from the cell interior, which is typically associated with internal vesicles. All pixel intensities at each pole were summed and divided by the total number of non-zero pixels. For visualization purposes, the segmented areas are denoted by red-dashed rectangles at the cell front and the rear (Supplementary Fig. 1b).

### Western blotting

Western blots were performed as previously described[10,22] using NuPage 4–12% Bis-Tris gels. Primary antibodies were applied at the following concentrations: anti-NHE1 (1:200; mouse, clone 54, Santa Cruz Biotechnology, sc-136239), anti-SWELL1 (1:100; mouse, clone 8H9, Santa Cruz Biotechnology, sc-517113) or GAPDH (1:1000; rabbit, clone 14C10, Cell Signaling Technology 2118), which was used as the loading control. Following overnight incubation with primary antibodies at 4 °C, membranes were washed 5× with TBST, and secondary antibodies were applied at 1:2000 dilution for 1 h at room temperature: anti-mouse IgG HRP-linked antibody (Cell Signaling Technologies) or anti-rabbit IgG HRP-linked antibody (Cell Signaling Technologies).

### Kaplan–Meier survival analysis

Analysis for breast cancer metastasis was performed using Kaplan–Meier plotter (https://kmplot.com). Auto scan mode was utilized to choose the cutoff between high and low expression cohorts.

### Treatment and inoculation of breast cancer cells in nude mice

Eight- to twelve-week-old female NOD.Cg-Prkdc < scid >/Jmice weighing 19–25 g were obtained from the University of Maryland at Baltimore and fed food and water ad libitum. The mice were maintained in accordance with the Institutional Animal Care and Use Committee procedures and guidelines of the University of Maryland at Baltimore. For subcutaneous injections, $1 \times 10^6$ luciferase/GFP-tagged MDA-MB-231 cells (SC, NHE1-KD, SWELL1-KD, or dual NHE1/SWELL1-KD) were suspended in 100 μL PBS and mixed with 25% of the total volume with Matrigel (Corning). Cell number was quantified via Countess® Automated Cell Counter (ThermoFisher), and confirmed by bioluminescent imaging. The cell suspension of SC or KD specimens was then injected subcutaneously into the fourth mammary gland on the ventral surface of the abdomen of the female mice in a blinded manner. Tumor volumes were measured by external caliper measurements weekly from the initial injection to the experimental endpoint. Tumors were measured along the two longest perpendicular axes in the x/y plane of each xenograft tumor to the nearest 0.1 mm with a digital caliper (Thomas Scientific, Inc.). Depth is assumed to be equivalent to the shortest of the perpendicular axes (y), and volume is calculated according to the: $V = xy^2/2$, as the standard practice for xenograft tumors. In accordance with the Institutional Animal Care and Use Committee procedures and guidelines of the University of Maryland at Baltimore, animals were restricted to a maximal tumor burden not to exceed 2 cm³. Mice bearing subcutaneous tumors were euthanized if tumors ulcerated, grew to 10% of the initial body weight, or reached 2 cm³. Signs of tumor ulceration or maximum tumor volume were recorded during each measurement. Tumor volume measurements were performed in a blinded manner.

### Bioluminescence imaging

Luciferase-expressing cells were injected subcutaneously into mice as above. At the indicated timepoints following injection, mice were injected intraperitoneally with D-luciferin potassium salt (150 mg/kg, Perkin Elmer) and returned to their cages for 5 min to allow for biodistribution. Mice were anesthetized with 2% isoflurane gas and imaged at 5 min intervals for the maximum photon emission. Total photon flux (photons/sec) was calculated and corrected for tissue depth by spectral imaging using Living Image 3.0 software (IVIS, Xenogen). Percent primary tumor growth was determined by subtracting the background from the peak signal during each measurement and normalizing it to the initial reading obtained for the same mouse.

Tissue samples collected at the time of necropsy were imaged for bioluminescence as described above in a blinded manner. Bioluminescence was only detected in viable cells expressing the firefly luciferase gene, indicative of an active metabolism. To avoid false positive

detection of bioluminescence signal, the background subtracted value was normalized to the background reading, and only readings that were ≥50x the background reading were considered to be positive for metastasis. To control for differences in tumor size, bioluminescence values were normalized to the volume of the primary tumor at the time of necropsy.

## Quantitative PCR

Quantitative PCR for human long interspersed nuclear elements (hLine) was conducted as previously described[12].

## Immunohistochemistry and pathology

Animals with primary tumor formation that exceeded the designated endpoint, including saturation exceeding 1000-fold over the initial bioluminescence signal, were sacrificed. Tissue samples were removed, fixed in formalin for 24 h, embedded in paraffin wax, and serially sectioned (4 μm thick). All immunohistochemistry GFP and H&E staining were performed by HistoWhiz (Brooklyn, NY).

## Ex Ovo chick embryo cancer xenograft model

Cancer cell extravasation assays were performed as described before[29,30], using 13-day-old fertilized White Leghorn chicken eggs acquired from the University of Alberta Poultry Research Centre. Briefly, $25–50 \times 10^3$ cancer cells were injected intravenously into the chicken CAM vein and allowed to extravasate for 8 h. Fifteen minutes before the assay CAM vasculature was visualized via injection of Lectin-649, and cancer cell extravasation was scored using intravital confocal imaging. At least seven animals were used for each condition for 3 experiments. All the procedures were approved by the University of Alberta Institutional Animal Care and Use Committee (IACUC).

## Statistics and reproducibility

All data represent the mean ± SEM or mean ± SD from ≥3 independent experiments (independent biological replicas) for each condition unless stated otherwise. The D'Agostino-Pearson omnibus normality test was used to determine whether data are normally distributed. Datasets with gaussian distributions were compared using Student's $t$-test (two-tailed) or one-way ANOVA followed by Tukey's post hoc test. For log-normal distribution, the statistical comparison was made after logarithmic transformation of the data followed by one-way ANOVA with post hoc Tukey. For comparing non-Gaussian distributions, the nonparametric Mann–Whitney U test or Kruskal–Wallis (with post hoc Dunn) were used for comparisons between two or more groups, respectively. Statistical significance was identified as $p < 0.05$. The exact $p$-values are provided in the Source Data file. Data were primarily collected and organized in Microsoft Excel (v. 15.30; Redmond, WA). Analysis was performed using GraphPad Prism 7.0b and 9.1.1 software (San Diego, CA).

In Fig. 1, images are representative of 4 (a) or 3 (c, g) or 2 (n) independent biological replicas. In Fig. 2, images are representative of 3 (b) or 4 (e, f) independent biological replicas. In Fig. 3, images are representative of 4 (a (i), (ii)) independent experiments. In Fig. 4, images are representative of 5 (a) or 6 (b) or 3 (f) independent experiments. In Fig. 5, images are representative of 3 (c, e) independent experiments. In Fig. 6, images are representative of 3 (h) independent experiments.

In Supplementary Fig. 1a, images are representative of 4 independent experiments. In Supplementary Fig. 4b, images are representative of 2 independent experiments. In Supplementary Fig. 5, images are representative of 4 (e) or 3 (h) or 3 (k, l) independent experiments. In Supplementary Fig. 6, images are representative of 2 (a) independent experiments.

## Theoretical Methods

**A multi-phase, steady-state cell migration model with charged ions-Mechanical part.** We used a multi-phase model[15,16] to understand the actin-water-ions-coupled cell migration. The model includes cytosol, F-actin, G-actin, and charged ions. A steady-state solution is sought. A confined cell in a channel can be modeled as a one-dimensional system. We use $x \in [0,L]$ to indicate the computational domain established in the moving frame of the cell, where $L$ is the cell length. The conservation of momentum and mass of the cytosol are

$$-\frac{dp}{dx} - \eta\theta_n(v_c - v_n) = 0, \frac{dv_c}{dx} = 0, \qquad (1)$$

where $p$ and $v_c$ are the hydraulic pressure and velocity of the cytosol, respectively; $\theta_n$ and $v_n$ are the concentration and velocity of the F-actin network, respectively; and $\eta$ is the coefficient of interfacial friction between the actin-network phase and the cytosol phase due to the velocity difference. The flux boundary condition for cytosol is

$$v_c - v_0 = -J_{\text{water}}^{\text{f}}, \text{at } x = L; v_c - v_0 = J_{\text{water}}^{\text{b}}, \text{at } x = 0, \qquad (2)$$

where $v_O$ is the steady-state velocity of the cell, $J_{\text{water}}$ is the water influx across the cell membrane, and the superscript 'f' and 'b' indicate quantities evaluated at the front and back end of the cell, respectively. Water flux is driven by the chemical potential difference of water across the cell membrane[44], and its expression is given by

$$J_{\text{water}}^{\text{f(b)}} = -\alpha^{\text{f(b)}}[(p^{\text{f(b)}} - p_*^{\text{f(b)}}) - RT(c^{\text{f(b)}} - c_0^{\text{f(b)}}] \qquad (3)$$

where $\alpha$ is the permeability coefficient of water, $c$ is the total concentration of all ion species, $R$ is the gas constant, and $T$ is the absolute temperature. In the model, we use the subscript '0' to indicate the extracellular environment. Due to hydraulic resistance, the hydraulic pressure exerted on the outside of the cell, $p_*$, is different from the hydraulic pressure at infinity, $p_0$. $p_*$ can be expressed as

$$p_*^{\text{f}} = p_0^{\text{f}} + d_g^{\text{f}}\left(v_0 - J_{\text{water}}^{\text{f}}\right), p_*^{\text{b}} = p_0^{\text{b}} - d_g^{\text{b}}\left(v_0 + J_{\text{water}}^{\text{b}}\right), \qquad (4)$$

where $d_g$ is the coefficient of external hydraulic resistance, which depends on the channel geometry and the viscosity of extracellular medium.

In this work, we do not explicitly model myosin contraction; we let the pressure in the actin network, $\sigma_n$, be dominated by passive pressure due to actin swelling. The constitutive relationship for the actin network is modeled as $\sigma_n = k_{\sigma_n}\theta_n$, where $k_{\sigma_n}$ is a constant. Focal adhesions provide forces to the cell through the actin network. These forces can be considered as an effective body force on the network. Therefore, the conservation of momentum of the actin network is written as

$$-\frac{d\sigma_n}{dx} + \eta\theta_n(v_c - v_n) - \eta_{\text{st}}\theta_n v_n = 0, \qquad (5)$$

where $\eta_{st}$ is the strength of focal adhesions. In the model, we allow actin polymerization to occur at the front of the cell, whereas depolymerization occurs throughout the cytoplasm. The mass conservation of the F-actin network and G-actin are

$$\frac{d}{dx}(v_n\theta_n) = -\gamma\theta_n, \frac{d}{dx}(v_c\theta_c) = D_{\theta_c}\frac{d^2\theta_c}{dx^2} + \gamma\theta_n, \qquad (6)$$

where $\theta_c$ and $D_{\theta_c}$ are the concentration and diffusion coefficient of G-actin, respectively. $\gamma$ is a constant rate of actin depolymerization. The

boundary condition for F-actin and G-actin are

$$\theta_n(v_0 - v_n) = J_{actin}, \theta_c(v_0 - v_c) = -J_{actin}, \qquad (7)$$

at the front of the cell, where $J_{actin} = J_{actin}^f \theta_c / (\theta_{c,c} + \theta_c)$ is the rate of actin polymerization; here $J_{actin}^f$ and $\theta_{c,c}$ are two constants. The actin flux is zero at the back of the cell. The total amount of actin is conserved such that $\int_0^l (\theta_n + \theta_c) dx = L\theta_*$, where $\theta_*$ is the average concentration of actin.

The cell experiences a frictional force with the channel wall. We let friction be proportional to cell velocity, i.e., $F_f = \xi v_0$, where $\xi$ is a friction coefficient, which applies to the negative direction of cell migration. Taken together, the force balance of the entire cell is

$$-\left(p_0^f - p_0^b\right) - \left(d_g^f + d_g^b\right)\left(v_0 - J_{water}^f\right) - \eta_{st}\int_0^L \theta_n v_n dx - F_f = 0 \qquad (8)$$

The system is solved by considering all the coupled equations together.

**Electrodynamics part.** Here we used a multi-species framework[45] to account for the electrodynamics part of the model. Species to consider include $Na^+$, $K^+$, $Cl^-$, $H^+$, $HCO_3^-$, $A^-$, $Buf^-$, and HBuf, where $A^-$ are intracellular impermeable charged proteins, $Buf^-$ is non-protonated solute, and HBuf is protonated solute in the buffer. We consider the following channels, pump, and transporters in the model: passive $Na^+$, passive $K^+$, passive $Cl^-$ (SWELL1), $Na^+/K^+$ pump (NKE), NHE, Anion Exchanger 2 (AE2). Combined with the mechanical part, the unknowns for the entire model are $p_c$, $v_c$, $v_n$, $\theta_n$, $\theta_c$, $c_{Na}$, $c_K$, $c_{Cl}$, pH, $c_A$, $c_{Buf}$, $\phi$, and $v_0$, where the $c$'s are solute concentrations having units of mM. The intracellular solute concentrations include $c_n = \{c_{Na}, c_K, c_{Cl}, c_H, c_{HCO_3}, c_A, c_{Buf}, c_{HBuf}\}^T$ and the extracellular solute concentrations include $c_n^0 = \{c_{Na}^0, c_K^0, c_{Cl}^0, c_H^0, c_{HCO_3}^0, c_A^0, c_G^0\}^T$.

The chemical equilibrium equation for the bicarbonate-carbonic acid pair is

$$CO_2(aq) + H_2O(l) \rightleftharpoons H^+(aq) + HCO_3^-(aq), \qquad (9)$$

where $[CO_2]_{aq}$ is related to the partial pressure of $CO_2$, $P_{CO_2}$, by the Henry constant $k_H$,

$$[CO_2]_{aq} = \frac{P_{CO_2}}{k_H}. \qquad (10)$$

The reaction equilibrium constant is

$$k_c = \frac{[HCO_3^-]_{aq}[H^+]_{aq}}{[CO_2]_{aq}}. \qquad (11)$$

Extracellular pH is defined as $pH_0 = -\log_{10}[H^+]_{aq,0}$ and $pK_c = -\log_{10}k_c$ so that Eq. 11 becomes

$$pH_0 - pK_c = \log_{10}\frac{[HCO_3^-]_{aq}^0}{P_{CO_2}/k_H}. \qquad (12)$$

$[CO_2]_{aq} = [CO_2]_{aq}^0$ since $CO_2$ can move freely across the cell membrane[46]. For the intracellular domain, we have

$$pH - pK_c = \log_{10}\frac{[HCO_3^-]_{aq}}{[CO_2]_{aq}}, \qquad (13)$$

where $pH = -\log_{10}[H^+]_{aq}$ is the intracelluar pH. The chemical reaction for the intracellular buffer solution is

$$HBuf(aq) \rightleftharpoons H^+(aq) + Buf^-(aq). \qquad (14)$$

The reaction equilibrium constant is similarly $k_B = [Buf^-]_{aq}[H^+]_{aq}/[HBuf]_{aq}$. With $pK_B = -\log_{10}k_B$, we obtain

$$pH - pK_B = \log_{10}\frac{[Buf^-]_{aq}}{[HBuf]_{aq}}. \qquad (15)$$

The flux for each species is

$$J_n = -D_n\frac{dc_n}{dx} + v_c c_n - D_n\frac{z_n F}{RT}c_n\frac{d\phi}{dx}, \qquad (16)$$

where $c_n$, $z_n$, and $D_n$ are the concentration, valance, and diffusion constant of each ion species, respectively. $\phi$ is the intracellular electrical potential. $F$, $R$, and $T$ are the Faraday's constant, ideal gas constant, and absolute temperature, respectively. The subscript '$n$' refers to different ion species, i.e., $n \in \{Na^+, K^+, Cl^-, H^+, HCO_3^-, A^-, Buf^-, HBuf\}$.

The governing equations and boundary conditions for $c_{Na}$, $c_K$, $c_{Cl}$, and $c_A$ are

$$-\frac{dJ_{Na}}{dx} = 0, J_{Na}|_{x=L} = -J_{Na}^f, J_{Na}|_{x=0} = J_{Na}^b, \qquad (17)$$

$$-\frac{dJ_K}{dx} = 0, J_K|_{x=L} = -J_K^f, J_K|_{x=0} = J_K^b, \qquad (18)$$

$$-\frac{dJ_{Cl}}{dx} = 0, J_{Cl}|_{x=L} = -J_{Cl}^f, J_{Cl}|_{x=0} = J_{Cl}^b, \qquad (19)$$

$$-\frac{dJ_A}{dx} = 0, J_A|_{x=L} = J_A|_{x=0} = 0, \qquad (20)$$

where the boundary fluxes directed inwards are defined as positive. The governing equation and boundary condition for pH are

$$-\frac{d}{dx}\left(J_{HCO_3} + J_{Buf} - J_H\right) = 0, \left(J_{HCO_3} + J_{Buf} - J_H\right)|_{x=L} = -\left(J_{HCO_3}^f + J_{Buf}^f - J_H^f\right),$$
$$\left(J_{HCO_3} + J_{Buf} - J_H\right)|_{x=0} = \left(J_{HCO_3}^b + J_{Buf}^b - J_H^b\right). \qquad (21)$$

where $J_{Buf}^{b/f} = 0$ due to the assumed non-permeability of buffer solutions. The governing equation and boundary condition for $c_{Buf}$ is

$$-\frac{d}{dx}\left(J_{Buf} + J_{HBuf}\right) = 0, \left(J_{Buf} + J_{HBuf}\right)|_{x=L} = \left(J_{Buf} + J_{HBuf}\right)|_{x=0} = 0. \qquad (22)$$

The total amount of non-permeable species are conserved such that

$$S\int_0^L c_A dx = N_A, S\int_0^L (c_{Buf} + c_{HBuf})dx = N_{Buf} + N_{HBuf}, \qquad (23)$$

where $S$ is the cross-sectional area of the cell. The derived qualities are

$$c_H = 10^3 10^{-pH}, c_{HCO_3} = \frac{P_{CO_2}}{k_H}10^{pH-pK_c}, c_{HBuf} = c_{Buf}10^{pK_B-pH}, \qquad (24)$$

which should be satisfied at all points in space. The intracellular electrical potential is solved by the electroneutrality condition, i.e., $\sum z_n C_n$.

The boundary flux for each ionic species is:

$$J_{Na}^{b/f} = J_{Na,p}^{b/f} + J_{NKE,Na}^{b/f} + J_{NHE,Na}^{b/f}, \tag{25}$$

$$J_K^{b/f} = J_{K,p}^{b/f} + J_{NKE,K}^{b/f}, \tag{26}$$

$$J_{Cl}^{b/f} = J_{Cl,p}^{b/f} + J_{AE2,Cl}^{b/f}, \tag{27}$$

$$J_H^{b/f} = J_{NHE,H}^{b/f}, \tag{28}$$

$$J_{HCO_3}^{b/f} = J_{AE2,HCO_3}^{b/f}. \tag{29}$$

For simplicity, in the notation below, we will omit the superscripts 'f/b' for membrane fluxes at the front and back of the cell.

The passive fluxes are tension-gated. We denote $G_m \in (0,1)$ as a mechanosensitive gating function that generally follows a Boltzmann distribution, i.e., $G_m = [1 + e^{-\beta_1(\tau_m - \beta_2)}]^{-1}$, where $\beta_1$ and $\beta_2$ are two constants and $\tau_m$ is the cortical/membrane tension, which can be calculated from the force balance at the membrane, i.e.,

$$\tau_m^f = \frac{b}{2}\left(\sigma_n^f + p_c^f - p_*^f\right), \tau_m^b = \frac{b}{2}\left(\sigma_n^b + p_c^b - p_*^b\right). \tag{30}$$

The passive ion fluxes, $J_{n,p}$, are proportional to the electrochemical potential difference of ions across the membrane[47],

$$J_{n,p} = \alpha_{n,p} G_m \left[RT\ln\Gamma_n - z_n F(\phi - \phi_0)\right], n \in \{Na^+, K^+, Cl^-\} \tag{31}$$

where $\Gamma_n = c_n^0/c_n$ is the ratio of extra- to intracellular ion concentrations; $\alpha_{n,p}$ is the permeability coefficient of each species, which depends on the channel property and the density of the channels in the membrane.

The Na$^+$/K$^+$ pump (NKE) is an active ion pump that maintains the membrane potential of cells. It exports three Na$^+$ ions and imports two K$^+$ ions per ATP molecule. Because the overall flux is positive outwards, the pump's activity depends on the membrane potential[48]. The NKE flux also depends on the concentrations of Na$^+$ and K$^+$ and saturates at high concentration limits[49]. Based on these facts, we model the flux of Na$^+$ and K$^+$ through the Na$^+$/K$^+$ pump as

$$J_{NKE} = J_{NKE,Na} = -\frac{3}{2}J_{NKE,K} = -\alpha_{NKE}G_{V,NKE}\left(1 + \beta_{NKE,Na}\Gamma_{Na}\right)^{-3}\left(1 + \beta_{NKE,K}/\Gamma_K\right)^{-2}, \tag{32}$$

where $\alpha_{NKE}$ is the permeability coefficient of the pump depending on the density of the pump as well as the concentration of ATP. $\beta_{NKE,Na}$ and $\beta_{NKE,K}$ are constants that scale $\Gamma_{Na}$ and $\Gamma_K$, respectively. The exponents 3 and 2 are Hill's coefficients of Na$^+$ and K$^+$, respectively. Equation 32 ensures that the flux is zero when either $1/\Gamma_{Na}$ or $\Gamma_K$ approaches zero; the flux saturates if $1/\Gamma_{Na}$ and $\Gamma_K$ approaches infinity. $G_{V,NKE}$ captures the voltage-dependence of the pump activity[48], $G_{V,NKE} = 2[1 + e^{-\beta_3(V_m - \beta_4)}]^{-1} - 1$, where $\beta_3$ and $\beta_4$ are constants.

The Na$^+$/H$^+$ exchanger (NHE), which has ten identified isoforms, is expressed in almost all tissues[46]. It imports one Na$^+$ and extrudes one H$^+$ under physiological conditions. This exchanger plays an important role in water flux, cell volume regulation[50] and cell migration[7]. NHE is quiescent at intracellular pH > 7.2[51]. The flux of NHE can thus be expressed as

$$J_{NHE} = J_{NHE,Na} = -J_{NHE,H} = \alpha_{NHE}G_{NHE}RT(\ln\Gamma_{Na} - \ln\Gamma_H), \tag{33}$$

where $\alpha_{NHE}$ is the permeability coefficient which does not significantly depend on cortical tension[52] and we assume it is constant. $G_{NHE} = [1 + e^{\beta_5(pH - \beta_6)}]^{-1}$ is a pH-gated function indicating the dependence of the NHE activity on pH.

The Cl$^-$/HCO$_3^-$ exchanger (AE2), which imports one Cl$^-$ and extrudes one HCO$_3^-$, is also common in cells. This exchanger is almost quiescent at intracellular pH < 6.8 − 7.3. Similarly, we assume that the flux takes the form

$$J_{AE2} = J_{AE2,Cl} = -J_{AE2,HCO_3} = \alpha_{AE2}G_{AE2}RT\left(\ln\Gamma_{Cl} - \ln\Gamma_{HCO_3}\right), \tag{34}$$

where $\alpha_{AE2}$ is the permeability coefficient of AE2 and is assumed to be independent of the cortical tension. $G_{AE2} = [1 + e^{-\beta_7(pH - \beta_8)}]^{-1}$ is a pH-gated function indicating the dependence of the AE2 activity on pH.

**Parameters.** The default parameters used in the model are listed in Supplementary Table 2. The model involves several degrees of freedom from the choice of parameters. The biophysical meaning of the degrees of freedom accounts for variations in cell types or different experimental conditions for the same cell line. For example, when NHE1 is inhibited, the corresponding parameter representing the NHE1 polarization ratio will change. The parameters that represent these degrees of freedom were fitted by Figs. 1e and 2a. Once all parameters were obtained, we used the parameters to predict cell velocity, including those presented in Fig. 3e and Supplementary Fig. 8.

The ratio of NHE1 polarization and SWELL1 polarization are among the most important parameters in the model because ion channel polarization is the underlying mechanism for the Osmotic Engine Model. Thus, we performed a parameter sensitivity study on ion channel polarization and other key parameters. For example, cell velocity is proportional to the ratio of NHE1 polarization, SWELL1 polarization, the rate of actin polymerization, and the strength of focal adhesions. Two contour plots indicating parameter dependence are shown in Supplementary Fig. 8.

### Reporting summary

Further information on research design is available in the Nature Research Reporting Summary linked to this article.

## Data availability

All data generated in this study are provided in the main manuscript, Supplementary Figs, and Source Data files. Patient distant metastasis-free survival (DMFS) time and gene expression data are available through KM-plotter[31]: https://kmplot.com/analysis/. All other relevant data supporting the key findings of this study are available from the corresponding authors upon reasonable request. Source data are provided with this paper.

## Code availability

The code for the two-phase model is available on GitHub in a link provided by SXS: https://github.com/sxslabjhu/ with https://doi.org/10.5281/zenodo.7105392[53].

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

## Acknowledgements

We thank Dr. Thomas J. Jentsch (Leibniz-Institut für Molekulare Pharmakologie (FMP), Berlin) for the generous gifts of SWELL1-GFP construct and anti-SWELL1 antibody, Dr. Antonio Frigeri (University of Bari) for the generous gift of AQP4-mCherry, Dr. Xavier Trepat (Institute for Bioengineering of Catalonia) for the kind gifts of ARHGEF11(DHPH)-Cry2-mCherry (OptoGEF), CAAX-CIBN-GFP and mito-CIBN-GFP as well as Dr. Heinrich Leonhardt (University of Munich) for the plasmids for LiPD system, pRing-CIBN-IR and pGBP-PHR-IR. We also thank Dr. Petr Kalab (Johns Hopkins) for insightful discussions. This work was supported, in part, by an NIH/NCI R01 CA254193 (K.K., S.S.M., and S.X.S.), R01 GM134542 (S.X.S and K.K.), NSF 2045715 (Y.L.), the Spanish Ministry of Science, Education and Universities through grants RTI2018-099718-B-100 (M.A.V.), an institutional "Maria de Maeztu" Programme for Units of Excellence in R&D and FEDER funds (M.A.V.), and postdoctoral fellowships from the Fonds de recherche du Quebec—Nature et technologies and the Natural Sciences and Engineering Research Council of Canada (A.K.). The opinions, findings, and conclusions, or recommendations expressed are those of the authors and do not necessarily reflect the views of any of the funding agencies.

## Author contributions

Y.Z. designed the study, performed most of the in vitro experiments, analyzed data, and wrote the manuscript. Y.L. and S.X.S. developed the mathematical models, provided critical input, and edited the manuscript. K.N.T. and S.S.M. performed and analyzed the in vivo mouse experiments, provided critical input, and edited the manuscript. K.S. and J.D.L. performed and analyzed the in vivo chick embryo experiments, provided critical input and edited the manuscript. Q.Y., K.B., S.J.L., R.Z., A.K., and Y.W. performed select in vitro experiments and analyzed data. P.M. designed (with Y.Z.) the optoSWELL1 constructs, and supervised the generation of various constructs used in this study, provided critical input, and edited the manuscript. S.A.S. and M.A.V. performed patch-clamp experiments, analyzed data, provided critical input, and edited the manuscript. K.K. designed and supervised the study and wrote the manuscript.

## Competing interests

The authors declare no competing interests.
