## [Peer Review File · Nature Communications]

REVIEWER COMMENTS

Reviewer #1 (Remarks to the Author):

The manuscript 'Polarized NHE1 and SWELL1 Regulate Migration Direction, Efficiency and Metastasis' by Yuqi Zhang et al. discusses the roles of SWELL1, a component of volume-regulated anion channel (VRAC) and NHE1, a cationic counterpart of SWELL1 for directed cell migration in confined spaces. The important roles of ion exchange regulation, in particular the interplay between volume increasing processes at the front and isosmotic shrinkage-inducing processes at the rear of a migrating cell, have been well documented and are quite nicely discussed in Schwab et al. *Physiol Rev* 92: 1865–1913, 2012. Whereas NHE1 has been known as a regulatory component for a long time, SWELL1 was identified somewhat recently (Qiu et al. *Cell*, Apr 10; 157(2): 447–458, 2014).

The present manuscript by Zhang et al., therefore, does not introduce novel molecular players or functionalities. It does, however, show new data on the localization of NHE1 and SWELL1 and very nicely demonstrate how interfering with their correct localization perturbs ion exchange, volume regulation and directed cell migration. The video data on the influence of NHE1/SWELL1 KD on cell movement out of a spheroid into a collagen matrix are also very impressive.

Moreover, the manuscript very nicely elucidates the role of Cdc42 on (re-)polarization.

However, given that experimental data already clearly show (without any modeling) that NHE1 and SWELL1 are involved in volume increase and shrinking at the leading and trailing edge, respectively, the additional information provided by the mathematical model is somewhat unclear.

Detailed comments:

The authors currently are showing rather few data on the localization of NHE and SWELL, in particular since their spatial distributions in Fig.1A are not as clear-cut as, for instance, that of PTEN in a fully polarized cell tends to be. It would be useful to add some supplementary figures showing NHE and SWELL distributions in migrating cells.

How was front/rear specified in the data shown in Fig. 1b?

Similar to the comment on the data showing the polarization of NHE1/SWELL1, I would suggest to show image data supporting Fig. 1D, which, interestingly, suggests that NHE1 and SWELL1 are really an interdependent pair (without additional similar components having somewhat redundant roles) since the double knockdown leads to a normal cell volume.

How does the effect of NHE1/SWELL1 depend on a particular composition of the extracellular medium?
What is the influence of changing the ion contents in the assays?

Fig. 2 makes it somewhat hard to assess the agreement between the mathematical model and the experimental data. Clearly, the authors could, in addition to showing the measured curves depicted in Fig. 2C, show how, experimentally, the velocity depends on the ratio of SWELL1 front to rear and also add error/variation bars to such a graph. It appears that relatively small changes in the distribution of SWELL1 are observed in cells reversing course (Fig. 2D). Could this be because the optically recruited SWELL1 is more effective due to its fixed membrane recruitment (absence of dynamic exchange with the cytoplasm)?

The model contains many parameters (suppl. Table 2) that are estimated or fitted. Given the rather sparse and phenomenological data, how were these fits/estimates made and how do these choices affect the behavior of the model? What do these degrees of freedom mean for the authors' interpretations that suggest, for instance, that the model 'predicts' the influence of SWELL1 polarization on migration in latrunculin treated cells (Fig. 2L)?

During the discussion of the data presented in Fig. 3, the authors state: "Because enrichment of SWELL1 expression at the cell rear confers migration directionality, and in light of the distinct polarization patterns of NHE1 and SWELL1 along the cell surface and their coordinated actions in confined cell migration, we hypothesized that efficient reversal of migration direction also requires NHE1 re-polarization to the new leading edge." Is the model able to reproduce this as well?

Movie S6 does not appear to show a reversal of migration direction, in contrast to the statement in the text (line 230).

Given the important role of Cdc42 for (re-)polarization of SWELL1 and NHE1 (and the necessity of also NHE1 depolarization for direction reversal), it is somewhat remarkable that the model (that does not include Cdc42) agrees so well with the data on directional reversal shown in Fig. 2. The authors could perhaps discuss this.

Reviewer #2 (Remarks to the Author):

The mechanisms that govern cell motility are not entirely understood; the Osmotic Engine Model proposes that cell movement is mediated by coordinated swelling at the cell leading edge and shrinkage at the trailing edge. In this manuscript the authors demonstrate that NHE1 and SWELL1 preferentially polarize at the cell leading and trailing edges, respectively, mediate cell volume regulation, and cell dissemination and migration. The investigators demonstrate that RhoA at the cell front triggers SWELL1 redistribution and Cdc42 controls NHE1 repolarization. The investigators further demonstrate that dual NHE1 and SWELL1 knockdown blocks cancer cell dissemination from spheroids and blocks extravasation and metastasis in vivo without impacting tumorigenesis. This is a well-executed and compelling study that builds a mechanistic model of how NHE1 and SWELL1 moderate motility. There is sufficient methodology that studies can be reproduced. The data supports the conclusions. The impact and significance of these studies is potentially high. This is a well crafted, innovative study.

There are some issues that should be addressed.

1. The investigators show that the inhibitory effects of combined NHE1 and SWELL1 depletion on migration/invasion are stronger than their individuals knock downs. In Figs. 4A, B the investigators show that NHE1-KD, SWELL1-KD, and NHE1/SWELL1-KD retard tumor growth equally in vivo. Given this, it is unclear why, in Fig. 4C, SWELL1-KD has a more significant impact on the number of LN, bone, and brain metastasis than combined NHE1/SWELL1-KD.
2. In Figure 4K depicts Kaplan-Meier analysis of NHE1 and SWELL1 expression; survival curves are very similar for SWELL1 vs. NHE1+SWELL expression – there is a lack of synergy.
3. In Figure 4K, it would be helpful to look at specific subtypes (Luminal A, Luminal B, HER2+, and TNBC). A gene can be preferentially associated with an aggressive subtype – e.g. Triple Negative Breast Cancer (TNBC) but not risk-stratify that subtype. It would be helpful to see if NHE1 and SWELL1, alone and together, can predict metastasis for TNBC. It would also be helpful to investigate in silico, the distribution of expression of NHE1 and SWELL1 in breast cancer sub-types (Luminal A, Luminal B, HER2+, and TNBC). A minor issue - the Y axis should be labeled metastasis-free survival (not survival).
4. Studies are for the most part performed in one cell line. The exception is for findings in Fig. 1 - where studies are replicated in SUM159 and PTEN^{-/-}/KRAS(G12V) MCF10A in supplementary data. It would be helpful for some of the other key in vitro modeling experiments to be confirmed in a second cell line.

Victoria Seewaldt, M.D.

Reviewer #3 (Remarks to the Author):

In this manuscript, Zhang and colleagues investigate the role of NHE1 and SWELL1 polarization in cancer cell migration. They find NHE1 localized to the front, and SWELL1 localized to the rear, of cells migrating in microchannels. Knockdown of NHE and Swell1 impacts cell volume and migration and diminishes cell disassociation from cancer cell spheroids. Polarization of SWELL1 impacts cell migration, and RhoA

activity regulates SWELL1 polarization, whereas CDC42 mediates migration reversal through NHE1 polarization. Finally, in a subcutaneous tumor model, they find that knockdown of both NHE1 and SWELL1 reduces breast cancer cell extravasation and metastasis.

Overall, this is a very nice manuscript that should be suitable for publication in Nature Communications following appropriate revisions. The authors very rigorously and elegantly show the roles of NHE1 and SWELL1 in cancer cell migration in channels, and their connection to RhoA and CDC42. These results are new for the field and add to the growing body of evidence that osmotic pressure and cell volume regulation play a critical role in cell migration. However, I have some concerns with some of the claims regarding the observed results and osmotic migration engine mechanism is overstated, and the connection of the polarized migration mode to 3D culture models and in vivo is less clear. The manuscript needs some additional revision experiments as well as some revisions of the claims prior to publication to address these concerns.

Major comments:

1. The generality of the specific migration mode observed in channels to non-channel contexts, such as collagen gels or the in vivo data is implied but not shown. Reduction of cell dissemination from spheroids and metastasis in a subcutaneous tumor with dual knockdown of SWELL1 and NHE is interesting, but that does not mean the same mechanism of migration. To show this, they would at least need to show the similar polarization of NHE1 and SWELL1 in those contexts. This would be straightforward in 3D, and they could do histology of the primary subcutaneous tumor in the in vivo studies to show this. Further, it seems the most relevant experiment in vitro in 3D collagen gels would be single cells cultured in collagen gels. The authors should conduct these experiments. If they don't show the same type of polarization, I don't think that takes away from the impact of the paper, but gives us a better idea of what contexts this type of migration is relevant to.

2. While it is assumed that the observed migration mode is exclusively using the OEM, it's not clear to me that that's the case from the data. For example, inhibition of actin polymerization with LatA in Fig. 2M diminishes cell migration, which differs from the observation from the authors' osmotic engine model paper (Stroka, et al., Cell 2014, Fig. 1A). Further, even with knockdown of both NHE1 and SWELL1, cells still do undergo migration (just slower) and metastases are observed (at lower levels). So, it appears that the migration might be somewhat different for these cells – perhaps some combination of actin and myosin mediated migration with the osmotic engine model? I think it would be helpful for the authors to clarify the role of actin and myosin early on, and conduct at least a select set of studies to confirm the osmotic engine mechanism is occurring from these cells (not everything from the Cell paper, but maybe 1 or 2 analyses), and include this early on. Finally, the authors should be clear in the abstract and text that this is more of a hybrid mode of migration (if this turns out to be the case). Again, I don't think this takes away from the impact of the paper but helps to understand it.

Minor comments:

1. Would be helpful to indicate why SWELL1 and AQP4, and not other related molecules were studied here.

Authors' Responses to Editor's and Reviewers' Remarks

First and foremost, we would like to thank the editor and the reviewers who have meticulously read our manuscript and provided us with insightful comments. We also greatly appreciate the invitation to submit a revised manuscript, which addresses all of the reviewers' concerns. Below, we have pasted each of the reviewers' comments followed by our responses. **Editorial changes or addition of new materials in our revised manuscript are shown in red.**

Reviewer #1

General comments: The manuscript 'Polarized NHE1 and SWELL1 Regulate Migration Direction, Efficiency and Metastasis' by Yuqi Zhang et al. discusses the roles of SWELL1, a component of volume-regulated anion channel (VRAC) and NHE1, a cationic counterpart of SWELL1 for directed cell migration in confined spaces. The important roles of ion exchange regulation, in particular the interplay between volume increasing processes at the front and isosmotic shrinkage-inducing processes at the rear of a migrating cell, have been well documented and are quite nicely discussed in Schwab et al. *Physiol Rev* 92: 1865–1913, 2012. Whereas NHE1 has been known as a regulatory component for a long time, SWELL1 was identified somewhat recently (Qiu et al. *Cell*, Apr 10; 157(2): 447–458, 2014).

The present manuscript by Zhang et al., therefore, does not introduce novel molecular players or functionalities. It does, however, show new data on the localization of NHE1 and SWELL1 and very nicely demonstrate how interfering with their correct localization perturbs ion exchange, volume regulation and directed cell migration. The video data on the influence of NHE1/SWELL1 KD on cell movement out of a spheroid into a collagen matrix are also very impressive.

Moreover, the manuscript very nicely elucidates the role of Cdc42 on (re-)polarization.

However, given that experimental data already clearly show (without any modeling) that NHE1 and SWELL1 are involved in volume increase and shrinking at the leading and trailing edge, respectively, the additional information provided by the mathematical model is somewhat unclear.

Authors' Response: We thank the reviewer for their critique, enthusiastic remarks and insightful comments, which have helped us to improve the quality of our manuscript. Regarding the mathematical model, we wish to respectfully point out that the model prompted us to test how the spatial localization of an ion channel can impact cell migration, which we next tested and verified experimentally.

Detailed comments:

Comment #1: The authors currently are showing rather few data on the localization of NHE and SWELL, in particular since their spatial distributions in Fig.1A are not as clear-cut as, for instance, that of PTEN in a fully polarized cell tends to be. It would be useful to add some supplementary figures showing NHE and SWELL distributions in migrating cells.

Authors' Response: Per reviewer's suggestion, we provide a set of an additional 4 representative confocal images showing the preferential enrichment of NHE1 and SWELL1 at the cell leading and trailing edges, respectively (**new Supplementary Fig. 1a**).

Comment #2: How was front/rear specified in the data shown in Fig. 1b?

Authors' Response: Having identified the cell directionality from the time-lapse recordings, we indicate in the newly attached schematic (**new Supplementary Fig. 1b**) the areas at the leading

and trailing edges of the pill-like cells that we considered to quantify the intensity of NHE1 and SWELL1.

Comment #3: Similar to the comment on the data showing the polarization of NHE1/SWELL1, I would suggest to show image data supporting Fig. 1D, which, interestingly, suggests that NHE1 and SWELL1 are really an interdependent pair (without additional similar components having somewhat redundant roles) since the double knockdown leads to a normal cell volume.

Authors' Response: We appreciate the reviewer's insightful remark. However, we wish to stress the heterogeneity in cell volume for scramble control (SC) and knockdown (KD) cells shown in Fig. 1D. As such, we will not do justice by selecting a few images that show visual differences. Nevertheless, we are providing a 3D reconstruction of a cell to demonstrate our method of cell volume quantification (**new Supplementary Fig. 1c**).

Comment #4: How does the effect of NHE1/SWELL1 depend on a particular composition of the extracellular medium? What is the influence of changing the ion contents in the assays?

Authors' Response: This is an excellent albeit open-ended question, which represents our future direction in this area. Nevertheless, following the reviewer's suggestion, we examined the influence of medium lacking Cl^- (which was substituted by glutamate) on confined cell migration. Our data reveal that cells, suspended either in regular DMEM-containing medium or in our self-assembled control medium containing chloride ions, display no difference in migration velocity inside confining channels. Interestingly, cells suspended in our self-assembled medium, in which chloride was substituted with glutamate (labeled as "modified"), exhibit lower velocity (**Fig. 1.1A**), which mirror our data with SWELL1-KD cells (see Fig. 1e in our manuscript).

In a separate set of experiments, cells in a sodium (Na^+)-free isotonic solution, prepared by replacing Na^+ with N-Methyl-D-glucamine, also display reduced cell velocity (**Fig. 1.1B**), as we observed with NHE1-KD cells (see Fig. 1e in our manuscript).

We chose not to include **Fig. 1.1** in our revised manuscript since we feel that these data are peripheral to the story of our manuscript. Ideally, we would like to include these data in a separate, systematic study in which we examine the effects of substitution of different ions, including Na^+ and Cl^- , on cell migration.

Comment #5: Fig. 2 makes it somewhat hard to assess the agreement between the mathematical model and the experimental data. Clearly, the authors could, in addition to showing the measured

curves depicted in Fig. 2C, show how, experimentally, the velocity depends on the ratio of SWELL1 front to rear and also add error/variation bars to such a graph. It appears that relatively small changes in the distribution of SWELL1 are observed in cells reversing course (Fig. 2D). Could this be because the optically recruited SWELL1 is more effective due to its fixed membrane recruitment (absence of dynamic exchange with the cytoplasm)?

Authors' Response: Regarding the reviewer's remark on the assessment of agreement between mathematical predictions and experimental data, we revised Fig. 2a, which now includes a) modeling predictions (shown in red), b) experimental measurements from 11 different cells subjected to optogenetic stimulation (shown in gray) and c) the average value \pm S.D. of the experimental data (shown in blue). **Revised Fig. 2a** shows an excellent agreement between mathematical predictions and experimental data.

Regarding the reviewer's remark on "relatively small changes" observed in Fig. 2d, we wish to respectfully point out that the average change of the front-to-rear SWELL1 intensity ratio is 2.4 fold. For visualization purposes and eliminate the potential confusion of the readers, we have extended the red dotted line, which represents the reference line for SWELL1 polarization (front-to-rear ratio = 1), as opposed to the black dotted line, which corresponds to zero velocity.

Comment #6: The model contains many parameters (suppl. Table 2) that are estimated or fitted. Given the rather sparse and phenomenological data, how were these fits/estimates made and how do these choices affect the behavior of the model? What do these degrees of freedom mean for the authors' interpretations that suggest, for instance, that the model 'predicts' the influence of SWELL1 polarization on migration in latrunculin treated cells (Fig. 2L)?

Authors' Response: We thank the reviewer for this important comment. The model does involve a number of degrees of freedom from the choice of parameters. We tested several combinations of parameters and found the best combination as reported in the parameter table. We have also performed parameter sensitivity study. For example, the cell velocity is roughly linearly proportional to the ratio of NHE polarization, SWELL1 polarization, the rate of actin polymerization, and the strength of focal adhesion. We have included two contour plot to indicate the parameter dependence (**Fig. 1.2**). The model was developed based on general physics principles underlying the cell migration problem. The biophysical meaning of the degrees of freedom accounts for

variations from different cell types or different experimental conditions for the same cell line. For example, when NHE is inhibited, the corresponding parameter representing the NHE polarization ratio will change. In our model, we find the parameters that fit the experimental data without Lat A to estimate the parameters for this particular cell type and confined experimental condition. We then use the parameters to predict the cell's response with Lat A treatment.

Comment #7: During the discussion of the data presented in Fig. 3, the authors state: "Because enrichment of SWELL1 expression at the cell rear confers migration directionality, and in light of the distinct polarization patterns of NHE1 and SWELL1 along the cell surface and their coordinated actions in confined cell migration, we hypothesized that efficient reversal of migration direction also requires NHE1 re-polarization to the new leading edge." Is the model able to reproduce this as well?

Authors' Response: The model does predict a reversal of cell migration when the NHE1 polarization is reversed as shown in **Fig. 2.3** below.

Comment #8: Movie S6 does not appear to show a reversal of migration direction, in contrast to the statement in the text (line 230).

Authors' Response: In this movie, the cell had already migrated from left (channel entrance) to right (channel exit). We applied the light stimulation at $t=10$ sec at the cell leading edge (yellow box), which resulted in the migration of cell from right to left, which we defined as reversal of migration direction. Unfortunately, we did not record the cell at earlier time points.

Comment #9: Given the important role of Cdc42 for (re-)polarization of SWELL1 and NHE1 (and the necessity of also NHE1 depolarization for direction reversal), it is somewhat remarkable that the model (that does not include Cdc42) agrees so well with the data on directional reversal shown in Fig. 2. The authors could perhaps discuss this.

Authors' Response: We thank the reviewer for this insightful comment. The model is capable of this prediction because it incorporates the combined effects of ion transport, water flux and actin dynamics. As such, a change of SWELL1 or NHE1 polarization leads to a change in the relevant ion fluxes, thereby modulating the intracellular ion composition and water flux. In sum, we believe the model captures the essential physical processes underlying the experimental observation.

Reviewer #2 Victoria Seewaldt, M.D.

General comments: The mechanisms that govern cell motility are not entirely understood; the Osmotic Engine Model proposes that cell movement is mediated by coordinated swelling at the cell leading edge and shrinkage at the trailing edge. In this manuscript the authors demonstrate that NHE1 and SWELL1 preferentially polarize at the cell leading and trailing edges, respectively, mediate cell volume regulation, and cell dissemination and migration. The investigators demonstrate that RhoA at the cell front triggers SWELL1 redistribution and Cdc42 controls NHE1 repolarization. The investigators further demonstrate that dual NHE1 and SWELL1 knockdown blocks cancer cell dissemination from spheroids and blocks extravasation and metastasis *in vivo* without impacting tumorigenesis. This is a well-executed and compelling study that builds a mechanistic model of how NHE1 and SWELL1 moderate motility. There is sufficient methodology that studies can be reproduced. The data supports the conclusions. The impact and significance of these studies is potentially high. This is a well crafted, innovative study.

Authors' Response: We thank the reviewer for their critique, enthusiastic remarks and insightful comments, which have helped us to improve the quality of our manuscript.

Detailed comments:

Comment #1: The investigators show that the inhibitory effects of combined NHE1 and SWELL1 depletion on migration/invasion are stronger than their individuals knock downs. In Figs. 4A, B the investigators show that NHE1-KD, SWELL1-KD, and NHE1/SWELL1-KD retard tumor growth equally *in vivo*. Given this, it is unclear why, in Fig. 4C, SWELL1-KD has a more significant impact on the number of LN, bone, and brain metastasis than combined NHE1/SWELL1-KD.

Authors' Response: First, please allow us to clarify that Fig. 4a,b show that silencing of NHE1 and/or SWELL1 does not alter primary tumor growth *in vivo*, whereas dual NHE1 and SWELL1 knockdown significantly suppresses metastasis to the lung and liver (Fig. 4e,f). Regarding the reviewer's remark on LN, bone and brain metastasis, we wish to respectfully point out that Fig. 4c reports the positive metastatic events in these organs irrespective of their size. However, as shown in Fig. 4e, the BLI levels of these metastatic tumors in LN, bone and brain are just above the background, and no statistically significant difference could be detected between any of these groups.

Comment #2: In Figure 4K depicts Kaplan-Meier analysis of NHE1 and SWELL1 expression; survival curves are very similar for SWELL1 vs. NHE1+SWELL expression – there is a lack of synergy.

Authors' Response: Cell migration *in vitro* using microfluidic assays and 3D spheroids recapitulate a few critical steps of the metastatic cascade, and as such *in vitro* data inform but do not necessarily mirror the responses of human subjects. We hope you agree with us that the jump from *in vitro* experiments to real life patients is gigantic. Having said this, the p value in the Kaplan-Meier analysis (Fig. 4k) for the dual NHE1 and SWELL1 expression ($p=0.00025$) is much smaller than that of SWELL1 alone ($p=0.0022$) suggesting a potential synergistic contribution.

Comment #3: In Figure 4K, it would be helpful to look at specific subtypes (Luminal A, Luminal B, HER2+, and TNBC). A gene can be preferentially associated with an aggressive subtype – e.g. Triple Negative Breast Cancer (TNBC) but not risk-stratify that subtype. It would be helpful to see if NHE1 and SWELL1, alone and together, can predict metastasis for TNBC. It would also be helpful to investigate *in silico*, the distribution of expression of NHE1 and SWELL1 in breast cancer

sub-types (Luminal A, Luminal B, HER2+, and TNBC). A minor issue - the Y axis should be labeled metastasis-free survival (not survival).

Authors' Response: We thank the reviewer for catching the mislabeling of the Y-axis in Fig. 4k, which has now been corrected in our revised manuscript.

Following the reviewer's suggestion, we have plotted in **Fig. 2.1A-D (please see p.7)** metastasis-free survival of different breast cancer subtypes (Luminal A, Luminal B, HER2+, and TNBC) followed by our brief remarks:

- i. Higher SWELL1 or higher dual NHE1 and SWELL1 expression correlates with lower metastasis-free survival in the aggressive TNBC breast cancer, which is in line with our *in vitro* and *in vivo* findings using TNBC cell lines (**Fig. 2.1D**). Similarly, high dual NHE1 and SWELL1 expression is associated with poorer prognosis in the luminal B (**Fig. 2.1B**), whereas no statistically-significant difference is observed for luminal A (**Fig. 2.1A**) or HER2+ (**Fig. 2.1C**) breast cancer patients (<https://kmplot.com/analysis/index.php?p=background>).
- ii. In general, KM plotter and TCGA data always need to be interpreted with caution. Below, we outline a few examples to support our claims:
 - a. **Higher NHE1 expression** in luminal B patients is associated with **better** metastasis-free survival and a hazard ratio, HR, of 0.64, whereas SWELL1 expression is not predictive of patient outcomes in these patients. Yet, **higher dual NHE1 and SWELL1 expression** in the luminal B patients correlates with **worse** metastasis-free survival and HR of 4.13. This analysis underlines the complexity of predicting clinical outcomes in dual NHE1 and SWELL1 expression data from their individual gene expression analysis.
 - b. We wish to stress **the markedly lower number of specimens in different subtypes** (e.g., ~80 for luminal B patients or 150 TNBC patients) relative to ~1000 patients for all breast cancer patients, **which may affect the accuracy of the findings**. Along these lines, the KM plotter website is updated frequently, and the values keep changing as the sample size increases. These changes are more pronounced for patient data obtained from a small size pool. As such, **we strongly feel that we should not include the analysis of different breast cancer subtypes in our manuscript due to their small size**.
- iii. Gene expression of NHE1 and SWELL1 in paired tumor and adjacent normal tissues are shown in **Fig. 2.1E and F (please see p. 8)**. NHE1 has increased expression in tumor relative to normal samples, whereas the opposite is true for SWELL1 (<https://tnmplot.com/analysis/>). However, these data do not provide a sufficient sample size to compare aggressive (metastatic) to non-metastatic tumors.

In sum, although we provide these data to the reviewer for her evaluation, we strongly feel that we should not include **Fig. 2.1** in our article **due to the small sample size of patients with different breast cancer subtypes**. Also, the NHE1 data discussed in **part iii** are in agreement with the findings presented in our manuscript, whereas others (**part i**) are not. On the other hand, the SWELL1 data discussed in **part iii** are not in agreement with the findings presented in our manuscript, whereas others (**part i**) are. **Importantly, these data fail to tell the most important aspect of our story that polarization of NHE1 and SWELL1 in distinct cellular compartments is responsible for the differential cell motility, which might be affected by motor proteins and processes, which are currently not fully characterized.**

Fig. 2.1A-D. Kaplan-Meier analysis of NHE1 expression (i), SWELL1 expression (ii), NHE1 and SWELL1 expression (iii) and metastasis-free survival for different breast cancer types.

Comment #4: Studies are for the most part performed in one cell line. The exception is for findings in Fig. 1 - where studies are replicated in SUM159 and PTEN^{-/-}/KRAS(G12V) MCF10A in supplementary data. It would be helpful for some of the other key in vitro modeling experiments to be confirmed in a second cell line.

Authors' Response: In our original submission, we demonstrated that NHE1 and SWELL1 inhibition suppressed confined migration in SUM159 and PTEN^{-/-}/KRAS(G12V) MCF10A cells. Per reviewer's suggestion, we performed the following key experiments with a second cell line:

- i. We verified the distinct polarization patterns of NHE1 and SWELL1 at the leading and trailing edge, respectively, of SUM159 cells, as evidenced by quantitative immunofluorescence analysis of their endogenous levels (**new Supplementary Fig. 1Si-iii**).
- ii. We verified the roles of NHE1 and SWELL1 in isosmotic swelling and shrinkage, respectively, in SUM159 cells (**new Supplementary Fig. 1T**).
- iii. We verified that single and combined inhibition of NHE1 and SWELL1 markedly suppressed cell dissociation from 3D spheroids (**new Supplementary Fig. 1Xi-iii**).

Reviewer #3

General comments: In this manuscript, Zhang and colleagues investigate the role of NHE1 and SWELL1 polarization in cancer cell migration. They find NHE1 localized to the front, and SWELL1 localized to the rear, of cells migrating in microchannels. Knockdown of NHE and Swell1 impacts cell volume and migration and diminishes cell disassociation from cancer cell spheroids. Polarization of SWELL1 impacts cell migration, and RhoA activity regulates SWELL1 polarization, whereas CDC42 mediates migration reversal through NHE1 polarization. Finally, in a subcutaneous tumor model, they find that knockdown of both NHE1 and SWELL1 reduces breast cancer cell extravasation and metastasis.

Overall, this is a very nice manuscript that should be suitable for publication in Nature Communications following appropriate revisions. The authors very rigorously and elegantly show the roles of NHE1 and SWELL1 in cancer cell migration in channels, and their connection to RhoA and CDC42. These results are new for the field and add to the growing body of evidence that osmotic pressure and cell volume regulation play a critical role in cell migration. However, I have some concerns with some of the claims regarding the observed results and osmotic migration engine mechanism is overstated, and the connection of the polarized migration mode to 3D culture models and *in vivo* is less clear. The manuscript needs some additional revision experiments as well as some revisions of the claims prior to publication to address these concerns.

Authors' Response: We thank the reviewer for their critique, enthusiastic remarks and insightful comments, which have helped us to improve the quality of our manuscript.

Detailed comments:

Comment #1: The generality of the specific migration mode observed in channels to non-channel contexts, such as collagen gels or the *in vivo* data is implied but not shown. Reduction of cell dissemination from spheroids and metastasis in a subcutaneous tumor with dual knockdown of SWELL1 and NHE is interesting, but that does not mean the same mechanism of migration. To show this, they would at least need to show the similar polarization of NHE1 and SWELL1 in those contexts. This would be straightforward in 3D, and they could do histology of the primary subcutaneous tumor in the *in vivo* studies to show this. Further, it seems the most relevant experiment *in vitro* in 3D collagen gels would be single cells cultured in collagen gels. The authors should conduct these experiments. If they don't show the same type of polarization, I don't think that takes away from the impact of the paper, but gives us a better idea of what contexts this type of migration is relevant to.

Authors' Response: The reviewer raises an insightful comment. We wish to respectfully point out that we herein demonstrate that NHE1 and SWELL1 polarization is dynamic and regulates cell motility. As such, in a primary subcutaneous tumor where cells are "immobile", we do not anticipate to detect any polarization. However, such a polarization pattern could be theoretically be detected during cell dissemination from a primary tumor. To test this, we examined the SWELL1 polarization pattern during dissociation of SWELL1-GFP-tagged cells from 3D spheroids embedded in a 3D collagen gel *in vitro*. **New Fig. 1n** shows that SWELL1 is preferentially localized at the cell trailing edge (identified by a white arrowhead) during cell dissociation from a 3D spheroid. Additional images verifying this polarization pattern of SWELL1 in dissociating cells from the same or other spheroids are shown in **new Supplementary Fig. 1y**. These new data validate and extend the significance of our findings inside confining channels.

Comment #2: While it is assumed that the observed migration mode is exclusively using the OEM, it's not clear to me that that's the case from the data. For example, inhibition of actin polymerization with LatA in Fig. 2M diminishes cell migration, which differs from the observation from the authors' osmotic engine model paper (Stroka, et al., *Cell* 2014, Fig. 1A). Further, even with knockdown of both NHE1 and SWELL1, cells still do undergo migration (just slower) and metastases are observed (at lower levels). So, it appears that the migration might be somewhat different for these cells – perhaps some combination of actin and myosin mediated migration with the osmotic engine model? I think it would be helpful for the authors to clarify the role of actin and myosin early on, and conduct at least a select set of studies to confirm the osmotic engine mechanism is occurring from these cells (not everything from the *Cell* paper, but maybe 1 or 2 analyses), and include this early on. Finally, the authors should be clear in the abstract and text that this is more of a hybrid mode of migration (if this turns out to be the case). Again, I don't think this takes away from the impact of the paper but helps to understand it.

Authors' Response: First and foremost, we completely agree with the reviewer's assessment regarding a hybrid mode of MDA-MB-231 cell migration. In line with our published data (Stroka et al., *Cell* 2014), MDA-MB-231 cells, treated with a high concentration (2 μ M) of LatA, which completely disrupts F-actin, continue to migrate inside narrow collagen I-coated channels, albeit with a reduced velocity (**Fig. 2m**). Confined cell locomotion is halted only upon LatA treatment of either NHE1-KD (Stroka et al., *Cell* 2014) or SWELL1-KD cells (**Fig. 2m**). Together, these data reveal the cooperative roles of ion transporters/channels and actin cytoskeleton in driving efficient MDA-MB-231 cell migration. A relevant statement was added to **Lines 226-227** of our revised manuscript. We also discuss about “the functional contribution of OEM to confined cell migration” in **Line 214** and the interplay between actomyosin cytoskeleton and OEM in **Lines 339-340**.

In sum, the goal of this manuscript was to extend the Osmotic Engine Model (OEM) of migration. Indeed, we achieved this through:

- i. the identification of SWELL1 as a key component of OEM;
- ii. the regulation of its spatiotemporal distribution via the development of novel optogenetics tools and its effect on cell directionality and migration
- iii. the discovery of the coordinated actions of NHE1 and SWELL1 in cell expansion and shrinkage, respectively.
- iv. the **contribution** of these critical OEM constituents to cell migration and metastasis *in vivo*.

Minor comments: Would be helpful to indicate why SWELL1 and AQP4, and not other related molecules were studied here.

Authors' Response: According to OEM, a cell migrating in a confining channel establishes a spatial gradient of ion transporters/channels and aquaporins (AQPs) in the cell membrane so that local **swelling at the leading edge** and **shrinkage at the trailing edge**, respectively, facilitate net cell movement. We previously reported that the Na⁺/H⁺ exchanger 1 (NHE1) and AQP5 polarize to the leading edge of breast cancer cells in confinement. In this manuscript, we show that NHE1 promotes isosmotic cell swelling consistent with its role in regulatory volume increase (RVI). As we state in **Lines 60-62** of our revised manuscript, we aimed to identify which ion channel and AQP govern the shrinkage or regulatory volume decrease (RVD) of the cell rear.

Previous work has established that **volume-regulated anion channels (VRACs) are key players in RVD process in vertebrate cells**. Although VRACs have been extensively studied, their molecular identity as LRRC8A or SWELL1 (1, 2) was discovered in 2014. As we indicate in our revised manuscript (please see **Lines 68-69**), we found that “SWELL1 mediates isosmotic

shrinkage consistent with its role in the local volume decrease”. Regarding the AQP partner of SWELL1, it was rather luck.

1. F. K. Voss *et al.*, Identification of LRRC8 heteromers as an essential component of the volume-regulated anion channel VRAC. *Science* **344**, 634-638 (2014).
2. Z. Qiu *et al.*, SWELL1, a plasma membrane protein, is an essential component of volume-regulated anion channel. *Cell* **157**, 447-458 (2014).

REVIEWER COMMENTS

Reviewer #1 (Remarks to the Author):

I have to admit that I am somewhat disappointed with the responses supplied by the authors as most of the issues regarding missing details or explanations have not been addressed but rather dismissed.

Comment #1: The authors currently are showing rather few data on the localization of NHE and SWELL, in particular since their spatial distributions in Fig.1A are not as clear-cut as, for instance, that of PTEN in a fully polarized cell tends to be. It would be useful to add some supplementary figures showing NHE and SWELL distributions in migrating cells.

Authors' Response: Per reviewer's suggestion, we provide a set of an additional 4 representative confocal images showing the preferential enrichment of NHE1 and SWELL1 at the cell leading and trailing edges, respectively (new Supplementary Fig. 1a).

Reviewer's response: The images added to the manuscript do not show a clear localization of SWELL in the rear, in contrast to the claims of the authors.

Comment #2: How was front/rear specified in the data shown in Fig. 1b?

Authors' Response: Having identified the cell directionality from the time-lapse recordings, we indicate in the newly attached schematic (new Supplementary Fig. 1b) the areas at the leading and trailing edges of the pill-like cells that we considered to quantify the intensity of NHE1 and SWELL1.

Reviewer's response: It would have been helpful to provide a description of how rear and front were determined on a cell-by-cell basis. A more detailed description could have mentioned, e.g., percentages of the cell length, or proportions of overall fluorescence along the long axis of the cell etc.. The cartoon provided in Fig. S Fig 1b is somewhat uninformative.

Comment #3: Similar to the comment on the data showing the polarization of NHE1/SWELL1, I would suggest to show image data supporting Fig. 1D, which, interestingly, suggests that NHE1 and SWELL1 are really an interdependent pair (without additional similar components having somewhat redundant roles) since the double knockdown leads to a normal cell volume.

Authors' Response: We appreciate the reviewer's insightful remark. However, we wish to stress the heterogeneity in cell volume for scramble control (SC) and knockdown (KD) cells shown in Fig. 1D. As such, we will not do justice by selecting a few images that show visual differences. Nevertheless, we are providing a 3D reconstruction of a cell to demonstrate our method of cell volume quantification (new Supplementary Fig. 1c).

Reviewer's response: Fig. 1D clearly claims that the NHE/SWELL double KO leads to a normal volume. If the figure shows these data, why can't the authors back the data up by showing images? The manuscript shows image data to corroborate a variety of assumed mechanisms. Why not for this?

Comment #4: How does the effect of NHE1/SWELL1 depend on a particular composition of the extracellular medium? What is the influence of changing the ion contents in the assays?

Authors' Response: This is an excellent albeit open-ended question, which represents our future direction in this area. Nevertheless, following the reviewer's suggestion, we examined the influence of medium lacking Cl⁻ (which was substituted by glutamate) on confined cell migration. Our data reveal that cells, suspended either in regular DMEM-containing medium or in our self-assembled control medium containing chloride ions, display no difference in migration velocity inside confining channels. Interestingly, cells suspended in our self-assembled medium, in which chloride was substituted with glutamate (labeled as "modified"), exhibit lower velocity (Fig. 1.1A), which mirror our data with SWELL1-KD cells (see Fig. 1e in our manuscript).

In a separate set of experiments, cells in a sodium (Na⁺)-free isotonic solution, prepared by replacing Na⁺ with N-Methyl-D-glucamine, also display reduced cell velocity (Fig. 1.1B), as we observed with NHE1-KD cells (see Fig. 1e in our manuscript).

We chose not to include Fig. 1.1. in our revised manuscript since we feel that these data are peripheral to the story of our manuscript. Ideally, we would like to include these data in a separate, systematic study in which we examine the effects of substitution of different ions, including Na⁺ and Cl⁻, on cell migration.

Reviewer's response: As this manuscript is about the influence of opposing ion channels on the migratory behavior of the cells, it would have been very instructive to include data showing the influence of the ion contents in the medium the cells are moving in.

Comment #5: Fig. 2 makes it somewhat hard to assess the agreement between the mathematical model and the experimental data. Clearly, the authors could, in addition to showing the measured curves depicted in Fig. 2C, show how, experimentally, the velocity depends on the ratio of SWELL1 front to rear and also add error/variation bars to such a graph. It appears that relatively small changes in the distribution of SWELL1 are observed in cells reversing course (Fig. 2D). Could this be because the

optically recruited SWELL1 is more effective due to its fixed membrane recruitment (absence of dynamic exchange with the cytoplasm)?

Authors' Response: Regarding the reviewer's remark on the assessment of agreement between mathematical predictions and experimental data, we revised Fig. 2a, which now includes a) modeling predictions (shown in red), b) experimental measurements from 11 different cells subjected to optogenetic stimulation (shown in gray) and c) the average value \pm S.D. of the experimental data (shown in blue). Revised Fig. 2a shows an excellent agreement between mathematical predictions and experimental data.

Regarding the reviewer's remark on "relatively small changes" observed in Fig. 2d, we wish to respectfully point out that the average change of the front-to-rear SWELL1 intensity ratio is 2.4 fold. For visualization purposes and eliminate the potential confusion of the readers, we have extended the red dotted line, which represents the reference line for SWELL1 polarization (front- to-rear ratio =1), as opposed to the black dotted line, which corresponds to zero velocity.

Reviewer's response: The 'After' part of Fig. 1D shows that six of the cells barely have a difference between front and back regarding SWELL1 distribution. Given also the images in S Fig. 1, the dynamic positioning of SWELL1 appears to play a less stringent role than suggested by the authors.

Comment #6: The model contains many parameters (suppl. Table 2) that are estimated or fitted. Given the rather sparse and phenomenological data, how were these fits/estimates made and how do these choices affect the behavior of the model? What do these degrees of freedom mean for the authors' interpretations that suggest, for instance, that the model 'predicts' the influence of SWELL1 polarization on migration in latrunculin treated cells (Fig. 2L)?

Authors' Response: We thank the reviewer for this important comment. The model does involve a number of degrees of freedom from the choice of parameters. We tested several combinations of parameters and found the best combination as reported in the parameter table. We have also performed parameter sensitivity study. For example, the cell velocity is roughly linearly proportional to the ratio of NHE polarization, SWELL1 polarization, the rate of actin polymerization, and the strength of focal adhesion. We have included two contour plot to indicate the parameter dependence (Fig. 1.2). The model was developed based on general physics principles underlying the cell migration problem. The biophysical meaning of the degrees of freedom accounts for variations from different cell types or different experimental conditions for the same cell line. For example, when NHE is inhibited, the corresponding parameter representing the NHE polarization ratio will change. In our model, we find the parameters that fit the experimental data without Lat A to estimate the parameters for this particular cell type and confined experimental condition. We then use the parameters to predict the cell's response with Lat A treatment.

Reviewer's response: Unfortunately, the authors' response does not explain how the quality of particular parameter sets was determined and whether model construction and parameter selection

were done before the model was used to predict, for instance, the influence of SWELL1 polarization on cell migration. It would be interesting for the readers to learn which experimentally observable variables were used to constrain the rather large parameter space. Without these explanations, the added value of the modeling remains unclear.

Comment #7: During the discussion of the data presented in Fig. 3, the authors state: "Because enrichment of SWELL1 expression at the cell rear confers migration directionality, and in light of the distinct polarization patterns of NHE1 and SWELL1 along the cell surface and their coordinated actions in confined cell migration, we hypothesized that efficient reversal of migration direction also requires NHE1 re-polarization to the new leading edge." Is the model able to reproduce this as well?

Authors' Response: The model does predict a reversal of cell migration when the NHE1 polarization is reversed as shown in Fig. 2.3 below.

Reviewer's response: The reversal in the model happens when the front to back ratio of NHE is greater than 1.5.

Comment #8: Movie S6 does not appear to show a reversal of migration direction, in contrast to the statement in the text (line 230).

Authors' Response: In this movie, the cell had already migrated from left (channel entrance) to right (channel exit). We applied the light stimulation at $t=10$ sec at the cell leading edge (yellow box), which resulted in the migration of cell from right to left, which we defined as reversal of migration direction. Unfortunately, we did not record the cell at earlier time points.

Reviewer's response: The manuscript claims that the movie shows reversal of migration. Yet, the authors claim that they, unfortunately, did not record the cell prior to its reversal. Are there not other, more complete, movies to support the manuscript's claims?

Comment #9: Given the important role of Cdc42 for (re-)polarization of SWELL1 and NHE1 (and the necessity of also NHE1 depolarization for direction reversal), it is somewhat remarkable that the model (that does not include Cdc42) agrees so well with the data on directional reversal shown in Fig. 2. The authors could perhaps discuss this.

Authors' Response: We thank the reviewer for this insightful comment. The model is capable of this prediction because it incorporates the combined effects of ion transport, water flux and actin dynamics. As such, a change of SWELL1 or NHE1 polarization leads to a change in the relevant ion fluxes, thereby modulating the intracellular ion composition and water flux. In sum, we believe the model captures the essential physical processes underlying the experimental observation.

Reviewer's response: The model includes many components. But not Cdc42. The data show that Cdc42 is very important. The authors should explain how the action of Cdc42 is - somehow - incorporated into their model.

Reviewer #2 (Remarks to the Author):

The authors have successfully responded to reviewer concerns.

Reviewer #3 (Remarks to the Author):

I am satisfied with the revised manuscript. I recommend publication.

AUTHORS' RESPONSES TO THE RESIDUAL COMMENTS OF REVIEWER #1

Below, we copied and pasted the reviewer's original comments, followed by our previous responses (in the first revision of our manuscript); the new reviewer's remarks and our follow-up responses marked in blue. We apologize for omitting details or explanations in our previous rebuttal. However, we have now clearly and meticulously addressed all the reviewer's points by providing the necessary additional information/clarification(s).

Reviewer #1 (Remarks to the Author):

I have to admit that I am somewhat disappointed with the responses supplied by the authors as most of the issues regarding missing details or explanations have not been addressed but rather dismissed.

Comment #1: The authors currently are showing rather few data on the localization of NHE and SWELL, in particular since their spatial distributions in Fig. 1A are not as clear-cut as, for instance, that of PTEN in a fully polarized cell tends to be. It would be useful to add some supplementary figures showing NHE and SWELL distributions in migrating cells.

Previous Authors' Response: Per reviewer's suggestion, we provide a set of an additional 4 representative confocal images showing the preferential enrichment of NHE1 and SWELL1 at the cell leading and trailing edges, respectively (new Supplementary Fig. 1a).

Reviewer's response: The images added to the manuscript do not show a clear localization of SWELL in the rear, in contrast to the claims of the authors.

Follow-up Authors' Response: We regret that the reviewer has missed our point, as in all images and videos shown in our manuscript there is a clear enrichment of SWELL1 localization at the rear (denoted by white arrowheads) relative to the front (denoted by yellow arrowheads) of cells migrating in confinement (please see below part of Suppl. Fig. S1a). To convey this point more clearly, we have now added the front to rear ratio of SWELL1-GFP signal intensity for each cell shown in Suppl. Fig. 1a. Details of intensity quantification are described below (please see our response to Comment #2).

This reviewer may have been distracted by the SWELL1-GFP signal at the "centre" of cells. Please note that when such large transmembrane proteins are overexpressed, they tend to

accumulate intracellularly. Importantly, in the case of SWELL1, it is known that in addition to its plasma membrane localization, it also appears in internal vesicles (which are the “dots” in the images), as described in Li et al PNAS 2020 (PMID: 33139539).

Comment #2: How was front/rear specified in the data shown in Fig. 1b?

Previous Authors' Response: Having identified the cell directionality from the time-lapse recordings, we indicate in the newly attached schematic (new Supplementary Fig. 1b) the areas at the leading and trailing edges of the pill-like cells that we considered to quantify the intensity of NHE1 and SWELL1.

Reviewer's response: It would have been helpful to provide a description of how rear and front were determined on a cell-by-cell basis. A more detailed description could have mentioned, e.g., percentages of the cell length, or proportions of overall fluorescence along the long axis of the cell etc.. The cartoon provided in Fig. S Fig 1b is somewhat uninformative.

Follow-up Authors' Response: Breast cancer cells, such as MDA-MB-231 and SUM159, migrate persistently inside confining microchannels, displaying a persistence index of >0.95! As such, and in clear contrast to random 2D migration, cell rear and front are readily evident, as cells migrate directionally and persistently from the microchannel inlet to the microchannel outlet. Nevertheless, cell directionality was also verified after inspection of time-lapse recordings. For quantification of NHE1 or SWELL1 polarization, a custom MATLAB script was used to segment fluorescence intensity at the cell front and rear, and exclude the signal from the cell interior, which is typically associated with internal vesicles. All pixel intensities at each pole were summed and divided by the total number of non-zero pixels. For visualization purposes, the segmented areas are denoted by red-dashed rectangles at the cell front and the rear, which extend an **average of ~10%** of the cell length from each of the poles. This material has now been added to our re-revised manuscript (Lines 642-646).

Supplementary Fig. 1b. Schematic (*left*) illustrating the cell region, denoted by red-dashed rectangles, that was segmented by a custom MATLAB script for intensity quantification of the front and rear of the confined cell (*right*) shown in (a,ii).

Comment #3: Similar to the comment on the data showing the polarization of NHE1/SWELL1, I would suggest to show image data supporting Fig. 1D, which, interestingly, suggests that NHE1 and SWELL1 are really an interdependent pair (without additional similar components having somewhat redundant roles) since the double knockdown leads to a normal cell volume.

Previous Authors' Response: We appreciate the reviewer's insightful remark. However, we wish to stress the heterogeneity in cell volume for scramble control (SC) and knockdown (KD) cells shown in Fig. 1D. As such, we will not do justice by selecting a few images that show visual differences. Nevertheless, we are providing a 3D reconstruction of a cell to demonstrate our method of cell volume quantification (new Supplementary Fig. 1c).

Reviewer's response: Fig. 1D clearly claims that the NHE/SWELL double KO leads to a normal volume. If the figure shows these data, why can't the authors back the data up by showing images? The manuscript shows image data to corroborate a variety of assumed mechanisms. Why not for this?

Follow-up Authors' Response: Again, we wish to stress the marked heterogeneity in cell volume for scramble control (SC) and knockdown (KD) cells, as shown in Fig. 1d. Yet, our conclusion was derived from the meticulous analysis of hundreds of cells. In response to the reviewer's remark, we include below representative images of SC and dual KD cells with volumes at the average values reported in Fig. 1d.

Comment #4: How does the effect of NHE1/SWELL1 depend on a particular composition of the extracellular medium? What is the influence of changing the ion contents in the assays?

Previous Authors' Response: This is an excellent albeit open-ended question, which represents our future direction in this area. Nevertheless, following the reviewer's suggestion, we examined the influence of medium lacking Cl⁻ (which was substituted by glutamate) on confined cell migration. Our data reveal that cells, suspended either in regular DMEM-containing medium or in our self-assembled control medium containing chloride ions, display no difference in migration velocity inside confining channels. Interestingly, cells suspended in our self-assembled medium, in which chloride was substituted with glutamate (labeled as "modified"), exhibit lower velocity (Fig. 1.1A), which mirror our data with SWELL1-KD cells (see Fig. 1e in our manuscript).

In a separate set of experiments, cells in a sodium (Na⁺)-free isotonic solution, prepared by replacing Na⁺ with N-Methyl-D-glucamine, also display reduced cell velocity (Fig. 1.1B), as we observed with NHE1-KD cells (see Fig. 1e in our manuscript). We chose not to include Fig. 1.1. in our revised manuscript since we feel that these data are peripheral to the story of our manuscript. Ideally, we would like to include these data in a

separate, systematic study in which we examine the effects of substitution of different ions, including Na⁺ and Cl⁻, on cell migration.

Reviewer's response: As this manuscript is about the influence of opposing ion channels on the migratory behavior of the cells, it would have been very instructive to include data showing the influence of the ion contents in the medium the cells are moving in.

Follow-up Authors' Response: We have added the relevant graphs (new Supplementary Fig. 1s, please see below) using Na⁺ and Cl⁻ free solutions into our re-revised manuscript, which mirror our results obtained with NHE1-KD and SWELL1-KD cells, respectively. Relevant text has been added into our re-revised manuscript (Lines 127-130 and Lines 503-522).

Comment #5: Fig. 2 makes it somewhat hard to assess the agreement between the mathematical model and the experimental data. Clearly, the authors could, in addition to showing the measured curves depicted in Fig. 2C, show how, experimentally, the velocity depends on the ratio of SWELL1 front to rear and also add error/variation bars to such a graph. It appears that relatively small changes in the distribution of SWELL1 are observed in cells reversing course (Fig. 2D). Could this be because the optically recruited SWELL1 is more effective due to its fixed membrane recruitment (absence of dynamic exchange with the cytoplasm)?

Previous Authors' Response: Regarding the reviewer's remark on the assessment of agreement between mathematical predictions and experimental data, we revised Fig. 2a, which now includes a) modeling predictions (shown in red), b) experimental measurements from 11 different cells subjected to optogenetic stimulation (shown in gray) and c) the average value ± S.D. of the experimental data (shown in blue). Revised Fig. 2a shows an excellent agreement between mathematical predictions and experimental data.

Regarding the reviewer's remark on "relatively small changes" observed in Fig. 2d, we wish to respectfully point out that the average change of the front-to-rear SWELL1 intensity ratio is 2.4 fold. For visualization purposes and eliminate the potential confusion of the readers, we have extended the red dotted line, which represents the reference line for SWELL1 polarization (front- to-rear ratio =1), as opposed to the black dotted line, which corresponds to zero velocity.

Reviewer's response: The 'After' part of Fig. 1D shows that six of the cells barely have a difference between front and back regarding SWELL1 distribution. Given also the images in S Fig. 1, the dynamic positioning of SWELL1 appears to play a less stringent role than suggested by the authors.

Follow-up Authors' Response: We thank the reviewer for the opportunity to stress our point in Fig. 2a (copied and pasted below), which plots the migration directionality (as denoted by positive or negative speeds) as a function of the front-to-rear ratio of SWELL1 expression. What is most relevant is that **the change of the ratio from <1** (before the optogenetic stimulation in Fig. 2d) **to >1** (after optogenetic stimulation) **determines migration direction**. Please also note that **cell motility halts at a ratio of 1** (Fig. 2a). Our quantitative analysis reveals that the relative fold change of front to rear SWELL1 intensity ratio is 2.72 in response to optogenetic stimulation (new Suppl. Fig. 2b). We note a range in the relative fold change, which is not surprising given the heterogeneity of breast cancer cell population. Importantly, we wish to stress that **migration speed scales with the degree of SWELL1 polarization** (Fig. 2a).

Comment #6: The model contains many parameters (suppl. Table 2) that are estimated or fitted. Given the rather sparse and phenomenological data, how were these fits/estimates made and how do these choices affect the behavior of the model? What do these degrees of freedom mean for the authors' interpretations that suggest, for instance, that the model 'predicts' the influence of SWELL1 polarization on migration in latrunculin treated cells (Fig. 2L)?

Previous Authors' Response: We thank the reviewer for this important comment. The model does involve a number of degrees of freedom from the choice of parameters. We tested several combinations of parameters and found the best combination as reported in the parameter table. We have also performed parameter sensitivity study. For example, the cell velocity is roughly linearly proportional to the ratio of NHE polarization, SWELL1 polarization, the rate of actin polymerization, and the strength of focal adhesion. We have included two contour plot to indicate the parameter dependence (Fig. 1.2). The model was developed based on general physics principles underlying the cell migration problem. The biophysical meaning of the degrees of freedom accounts for variations from different cell types or different experimental conditions for the same cell line. For example, when NHE is inhibited, the corresponding parameter representing the NHE polarization ratio will change. In our model, we find the parameters that fit the experimental data without Lat A to estimate the parameters

for this particular cell type and confined experimental condition. We then use the parameters to predict the cell's response with Lat A treatment.

Reviewer's response: Unfortunately, the authors' response does not explain how the quality of particular parameter sets was determined and whether model construction and parameter selection were done before the model was used to predict, for instance, the influence of SWELL1 polarization on cell migration. It would be interesting for the readers to learn which experimentally observable variables were used to constrain the rather large parameter space. Without these explanations, the added value of the modeling remains unclear.

Follow-up Authors' Response: The fundamental idea of OEM is that ion channel polarization is needed for efficient cell migration. A symmetric cell cannot move, as noted for cells with equal SWELL1 expression (i.e., signal intensity levels) at both poles. The model does involve several degrees of freedom from the choice of parameters. Figures 1e and 2a (of our revised manuscript) were used to fit the parameters. The other results (including new Suppl. Fig. 4 below, which was included in our previous rebuttal letter) were model predictions. We have also performed a parameter sensitivity study. For example, cell velocity is proportional to the ratio of NHE1 polarization, SWELL1 polarization, the rate of actin polymerization, and the strength of focal adhesions. We have included two contour plots to indicate the parameter dependence (new Supplementary Fig. 4). Also, relevant text has been added into our revised manuscript (Lines 861-874).

The model was developed based on general physics principles underlying the cell migration problem. The biophysical meaning of the degrees of freedom accounts for variations from different cell types or different experimental conditions for the same cell line. For example, when NHE1 is inhibited, the corresponding parameter representing the NHE1 polarization ratio will change. In our model, we find the parameters that fit the experimental data without Lat A to estimate the parameters for this particular cell type and confined experimental condition. We then use the parameters to predict the cell's response with Lat A treatment.

Comment #7: During the discussion of the data presented in Fig. 3, the authors state: "Because enrichment of SWELL1 expression at the cell rear confers migration directionality, and in light of the distinct polarization patterns of NHE1 and SWELL1 along the cell surface and their coordinated actions in confined cell migration, we hypothesized that efficient reversal

of migration direction also requires NHE1 re-polarization to the new leading edge." Is the model able to reproduce this as well?

Previous Authors' Response: The model does predict a reversal of cell migration when the NHE1 polarization is reversed as shown in Fig. 2.3 below.

Reviewer's response: The reversal in the model happens when the front to back ratio of NHE1 is greater than 1.5.

Follow-up Authors' Response: Our model captures the correct *trend* of cell response and reflects the correct mechanism that we verified experimentally and presented in our manuscript. It is not surprising to see cells stalling or reversing at a ~1.5 front-to-back NHE1 ratio. We wish to respectfully emphasize that "*SWELL1 polarization confers migration direction*" (see line 35). As shown in Fig. 3m, optogenetic enrichment of SWELL1 at the old leading edge in the presence of the Cdc42 inhibitor, ML141, is sufficient to reverse migration direction. Yet, ML141 treatment markedly suppressed the migration velocity after cells reversed direction, which is attributed to the lack of NHE1 repolarization in Cdc42-inhibited cells (Fig. 3k, m, n). As shown in Fig. 3k, the front to rear ratio of NHE1 expression is greater than 1.5 (almost 2.0).

Comment #8: Movie S6 does not appear to show a reversal of migration direction, in contrast to the statement in the text (line 230).

Previous Authors' Response: In this movie, the cell had already migrated from left (channel entrance) to right (channel exit). We applied the light stimulation at t=10 sec at the cell leading edge (yellow box), which resulted in the migration of cell from right to left, which we defined as reversal of migration direction. Unfortunately, we did not record the cell at earlier time points.

Reviewer's response: The manuscript claims that the movie shows reversal of migration. Yet, the authors claim that they, unfortunately, did not record the cell prior to its reversal. Are there not other, more complete, movies to support the manuscript's claims?

Follow-up Authors' Response: Unfortunately, this set of experiments (and thus video recording) was performed by a student who has already graduated. Her rationale was that since the cell had entered the channel and given its highly persistent migration in microchannels (with an index of 0.98!), recording of earlier timepoints was not necessary. Given the reviewer's remark, we decided to omit this video from our re-revised manuscript.

Comment #9: Given the important role of Cdc42 for (re-)polarization of SWELL1 and NHE1 (and the necessity of also NHE1 depolarization for direction reversal), it is somewhat remarkable that the model (that does not include Cdc42) agrees so well with the data on directional reversal shown in Fig. 2. The authors could perhaps discuss this.

Previous Authors' Response: We thank the reviewer for this insightful comment. The model is capable of this prediction because it incorporates the combined effects of ion transport, water flux and actin dynamics. As such, a change of SWELL1 or NHE1 polarization leads to a change in the relevant ion fluxes, thereby modulating the intracellular ion composition and water flux. In sum, we believe the model captures the essential physical processes underlying the experimental observation.

Reviewer's response: The model includes many components. But not Cdc42. The data show that Cdc42 is very important. The authors should explain how the action of Cdc42 is - somehow - incorporated into their model.

Follow-up Authors' Response: We thank the reviewer for giving us the opportunity to further clarify this point. As shown experimentally, Cdc42 facilitates the repolarization of NHE1. If we were to model the *process* of repolarization (which is out of the scope of the current modeling goal), we would include Cdc42 in the model. However, our model does not focus on how a cell is repolarized. Instead, it encompasses the result of repolarization, which is manifested by the re-distribution of the ion transporter at the front and back of the cells. In other words, the model takes the channel distribution ratio as an input to predict migration speed and directionality. Thus, the effect of Cdc42 is implicitly incorporated into the model as a parameter of the channel distribution ratio.

In closing, we wish to emphasize that the mathematical modeling represents a very minor component of this manuscript as evidenced by the fact that only 3 out of ~100 figure panels encompass modeling data. Yet, the math modeling provided the initial insight that ion channel polarization controls migration directionality and also reinforced our experimental findings. This manuscript is clearly experiment-based, and includes the development of novel optogenetic tools for SWELL1, the use of sophisticated microfluidic and 3D spheroid assays, coupled with electrophysiology studies and two distinct animal models (mouse and chick embryo).

REVIEWERS' COMMENTS

Reviewer #1 (Remarks to the Author):

The authors have satisfyingly addressed most of the issues I raised and, given the overall scope of this massive project, I am fine with the re-revised version of the manuscript.